# Modeling Barcelona sidewalks: A high resolution urban scale assessment of the geometric attributes of the walkable network

**Francesc Valls** [1], **Álvaro Clua** [2]*

1 Department of Architectural Representation, UPC-BarcelonaTech, Barcelona, Spain, 2 Barcelona Laboratory of Urbanism, Department of Urban Design and Territorial Planning, UPC-BarcelonaTech, Barcelona, Spain

☯ These authors contributed equally to this work.
* alvaro.clua@upc.edu

**Data Availability Statement:** All relevant data are within the paper and its Supporting information files.

**Funding:** The author(s) received no specific funding for this work.

## Abstract

Within the framework of urban pedestrian networks, this paper presents a method of modeling the street network from the perspective of foot traffic, beyond the vehicle-focused street centerline representation approach in transportation research. A scalable method to extract the centerlines of the complete walkable urban area from its polygon representation at a one-meter resolution is discussed, using open-source tools. To evaluate the betweenness centrality in a spatially directed graph, the process is applied to the study of the 'walkable Barcelona', focusing on three key parameters: 1) the street width, 2) the longitudinal slope, and 3) the crosswalks connecting the sidewalk platforms. The results identify the uneven distribution of these parameters within a complex urban fabric, and the high-resolution cartography allows the identification of critical areas within the network, introducing future lines of research and applications of the workflow. This is especially relevant considering the increasing awareness of citizens and the urban agendas worldwide, aimed at improving and widening the sidewalk infrastructure that supports local activity in cities.

## 1. Introduction

### 1.1. Background

Mobility on foot is not only the most usual mode of transport in cities but also the basic condition for effective social interaction and activating commercial activity on ground floors. Walking is the essential action that prompted the foundation of historic cities but is currently a key asset for achieving more sustainable, livable, and vibrant environments. The importance of walking is also becoming especially relevant regarding citizens' awareness and political agenda in many cities around the world, advocating for proximity as a key urban value and a more active way of living. However, although there is rich urban knowledge about the infrastructures and criteria that facilitate the transit of motorized vehicles, cyclists, or public transport, pedestrian infrastructures have oftentimes been neglected in urban research, despite being the most equal, sustainable, and healthy way to move around the city.

**Competing interests:** The authors have declared that no competing interests exist.

In the last few years, the concern for walking in cities is getting increased interest, mainly due to the need for healthy mobility and social distancing during the COVID-19 restrictions and urban sustainability awareness. This has prompted the development of multiple tactical or permanent urban transformations of pedestrian areas in many cities worldwide. However, these transformations aimed at providing more space for pedestrians and adjusting the role of private motorized transport in cities, are often carried out following circumstantial opportunities or an even distribution of improvement of the public space throughout the city. In many cases, the result is exemplary and most of the solutions are becoming part of the know-how of redesigning streets available in a number of street-design catalogs such as the *National Association of City Transportation Officials* (https://nacto.org/ *Accessed 13 February 2023)* or the *Boston Complete Streets Guidelines (*https://bostoncompletestreets.org/ *Accessed 13 February 2023)*, among many others. Nevertheless, it might be stated that a deep evidence-based, and holistic approach to walking in cities as a mode of transport and as a system is still needed.

From the urban research sphere, this concern about walkable spaces is being addressed from multiple points of view. Starting from well-known essays such as *Walkable city rules* [1] to other studies in the Spanish context such as *La ciudad paseable* [2] or *A pie o en bici* [3], the key themes in order to take back cities for pedestrians are being described and thoroughly structured. At the same time, a large number of researchers are making great advances in the concept of 'walkability', focusing on understanding and parametrizing the attributes which may explain the reasons for usual walking itineraries [4–8].

'Walkability' has been also addressed from the perspective of spatial network analysis, with relevant research on the prediction of pedestrian flows using data analysis and studies of shortest itineraries. Among them, it is worth highlighting those recently elaborated by the MIT City Form Lab, through the development of the Urban Networks Analysis Toolbox [9], the pedestrian choice prediction works based on the study of attraction variables and GPS tracking [10, 11], or the use of image recognition algorithms to understand the impact of frontages in walking itineraries [12]. In a similar line are also works such as *Desirable streets* developed by the Senseable City Lab MIT, researching the factors that lead to changing minimum walking routes for larger distances but greater attraction [13]. Also noteworthy are projects such as the *Healthy Street Index* (https://www.underscorestreets.com/the-healthy-streets-index *Accessed 13 February 2023)* in which a distinction has been made between sections of London streets according to a health index that includes factors such as air quality, accessibility, walkability, and green spaces. Again, in the Spanish context, some relevant studies in this field are also digging into the influence of density and activities on pedestrian networks in Madrid [14, 15] or on the city of Toledo [16].

However, it should be noted that, apart from some specific works directed by A. Sevtsuk [17], the mapping and modeling unit of the previous research is the axial line of the street and not the exact description of the sidewalks as independent platforms interconnected by crosswalks. Although this simplification is sometimes argued by the inexistence of available cartographies of the sidewalks, this limitation should be considered when talking about walkability, as it necessarily implies a higher resolution in the cartography and detail in their interpretation.

This paper aims to present the results of the first steps of an in-progress research avenue focusing on the empirical evaluation of the walkable urban network [18], by providing and describing a method for modeling the sidewalks and pedestrian areas taking into account their essential geometrical attributes: the width, the slope and the topology of the network. The method is applied to the analysis of Barcelona and adjacent contiguous urban fabrics.

## 1.2. The sidewalks as a field for urban studies

The research is centered on the study of sidewalks as the essential infrastructure for urban walking [19]. These platforms, widely developed since the 18th Century as a safeguarded paved space separated from traffic and dust, are a prominent element in most cities today. Sidewalks might be regarded as a true urban system, and, to a large extent, it remains segregated from other modes of transport. The transfer from one sidewalk to another is done through crosswalks, which work as connection bridges—often controlled by traffic lights— and, therefore, substantially increase the walking times of the itineraries. Generally speaking, sidewalks are generally located at the sides of the streets, because these are the access platforms for shops and dwellings. Sometimes the sidewalks occupy the full section of the street, from façade to façade, thus configuring a pedestrian or shared street, more or less subdivided by several urbanization elements and with a greater or lesser degree of coexistence with road traffic.

The attention to sidewalks as an independent *urban system* has been the object of recent studies in some cities. Among them, is it worth highlighting *Sidewalks Widths* (M. Harvey) for the city of New York, where an interactive map categorizes the sidewalks of the city according to their width and according to the social distance derived from the pandemic (https://www.sidewalkwidths.nyc/ *Accessed 13 February 2023*). Similarly, works such as the *Milan sidewalks* map [20] are aimed at providing a detailed sidewalks map and, again, their categorization by the available width. At the Spanish level, studies carried out by SIGTE (Univ. Girona) on the mapping of sidewalks in Girona are also notable. However, although these works are considered a key reference for the research, they are not yet considering a deep systematic reading of the sidewalks network, as they do not include those crosswalks which connect sidewalks or the public spaces that are also accessible to the public. Only recent works on the Seattle sidewalk network using data retrieved from *OpenSidewalks Project* are relevant exceptions on this issue [21].

## 1.3. Literature review on modeling sidewalks

The challenge of studying the sidewalks as a system requires preliminary work of modeling the sidewalks and pedestrian spaces. This process starts with the definition of a polygon, the extraction of the centerline, and the computation of the essential geometrical attributes: the width, the slope, and the continuity. In the literature, the SFCGAL software library has been used to extract the skeleton network from a polygonal geometry, although it has been mainly applied to extract the indoor network from building data [22]. Another approach focuses on OpenStreetMap data to extract the sidewalk geometries; the OSM Sidewalkreator QGIS plugin [23] can approximate this task with an intuitive user interface but is currently limited to small areas.

Secondly, the accurate determination of the width of the sidewalks is a complex problem that has been approached from multiple methodological perspectives; in the case of the city of Vienna (Austria), three geometry-based methods were applied to estimate the sidewalk widths [24], two based on the definition of circles (inscribed or circumscribed), and another based on the area occupied by the sidewalks, obtaining considerable differences in the results depending on the method.

Moreover, the width calculation requires the determination of the centerline from which it is calculated, but the application of these methods to very large and complex areas such as the pedestrian network of a complete city is challenging; while it is feasible to produce these centerlines manually for small areas of study, for larger areas it is necessary to use automated tools and has been applied for map generalization purposes [25], and to extract the centerlines of watercourses from their polygon representation [26].

Finally, regarding the modeling of slopes in urban networks, most studies focus on walkability scores [27] or pedestrian catchment areas [28], using data from contour maps, or digital elevation models. The results of these studies in Sydney (Australia) and Milton (United Kingdom) reveal the importance of considering slope in pedestrian modeling and considering both slope magnitude and uphill and downhill situations in network analysis.

### 1.4. 'Walkable Barcelona' as a case study

This work is focused on modeling the pedestrian network of the so-called 'walkable Barcelona' on September 1, 2022, an area defined by the smooth pedestrian continuity or by those barriers that mark a clear inflection in daily walking itineraries. The result is an area of around 9,138 Ha, defined by the river Besòs, the Collserola mountain range (next to Ronda de Dalt ring road), the Mediterranean sea, and the border formed by the river Llobregat and the southern Ronda de Dalt (Fig 1). This area includes not only the municipality of Barcelona (except for the Vallvidrera nucleus, at the top of Collserola, and the restricted area of the Port of Barcelona) but also the municipality of L'Hospitalet de Llobregat, the western part of Sant Adrià del Besòs and some specific areas of Esplugues de Llobregat and Cornellà de Llobregat. According to Census Data provided by the Municipalities in 2022, the studied area has 1,936,123 inhabitants.

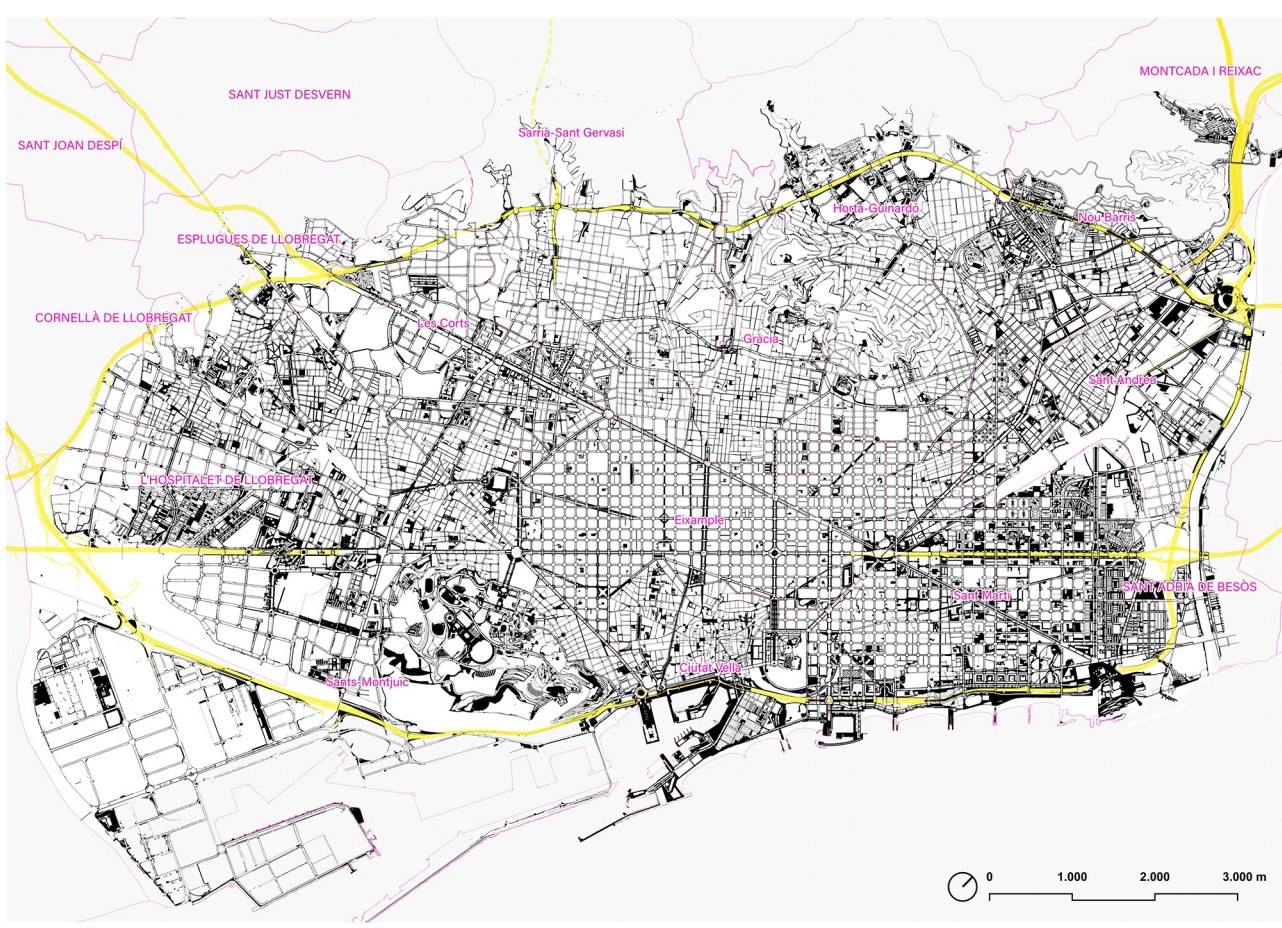

**Fig 1. Map of the sidewalks and pedestrian areas of the 'walkable Barcelona'.** In yellow, Barcelona's Rondes ring road. Source: the authors. Base Cartography from CartoBCN, under CC BY 4.0.

The interest in using Barcelona as a case study for this research is partly due to the close knowledge of the city from the author s' perspective but, above all, because of the recognition of Barcelona's public space in the urban debate. Public space care policies strongly started on the verge of the Olympic Games in '92 [29–33] and became the key element of what has been called the "Barcelona model" [30, 34–40]. The collections on monographs published by the Barcelona City Council, the Agència d'Ecologia Urbana de Barcelona—Barcelona Regional [41] and the Barcelona Metropolitan Area on the occasion of the upcoming Metropolitan Urban Master Plan (PDU), provide a clear example of the strategies and the design value that are guiding the construction of the Barcelona metropolis under this perspective.

Barcelona's street network has also been a subject of discussion in recent research and studies. Among them, it is worth mentioning the "Mercè" project (300,000km/s) where data from the training of AI algorithms for the recognition of Google Street Maps images has been joined with accessibility, sociodemographic, and spatial data at the street section (http://merce. 300000.eu/ *Accessed 13 February 2023*). Along the same line, it is worth highlighting some aspects of the "Aire / Air" project (https://air.300000.eu *Accessed 13 February 2023*), on air quality in Barcelona and, more specifically, the differentiation of street sections according to this variable. Also noteworthy are the studies on the visual perception of public spaces in central Barcelona using *visual graph analysis* [42] or other works digging into the relationship between the spatial configuration of Ciutat Vella and the acoustics from music street performances [43]. Finally, the modeling of the network has also been the basis of a number of recent Barcelona-centered research, including studies on the distribution of proximity retail [44]; the location of terraces of restaurants and bars in public spaces [45], or social media in public spaces [46].

Walking as a mode of transport in Barcelona also has been the object of specific studies on sustainable mobility developed by the IERMB (UAB), regarding statistical and metropolitan-scale studies on issues such as mobility of the elderly [47], urban networks for cyclists, mobility pedestrian proximity [48] or the studies on sustainable metropolitan mobility (Pla Metropolità de Mobilitat Urbana PMMU-AMB-IERMB, 2019–2024).

However, as a final consideration, it is important to observe that most of these studies have the ultimate goal of understanding the reasons for the pedestrian movement and the multiple conditions that finally influence the decision-making process, rather than the understanding of the sidewalks as infrastructures for mobility. There is indeed no extensive knowledge about the sidewalk infrastructure of Barcelona as an object of study itself, as a transport platform with its internal laws. If hazardous pedestrian itineraries are not considered now and walking is viewed as a means of transport, a thorough discussion on efficiency based on minimum distances, continuity, environmental quality or safety might be already proposed.

## 2. Materials

### 2.1. Source data

The geometry of the sidewalk polygon is based on raw data retrieved from three ancillary databases. For modeling Barcelona municipal area, the available cartography from CartoBCN (https://w20.bcn.cat/cartobcn/ *Accessed 2 September 2022)* repository was used. This database called *Municipal Topographic Cartography* is a GeoPackage file and offers comprehensive high-resolution cartography of the map of Barcelona (2021). For this research, only some geometries were taken into consideration: 1) the geometry of the urban blocks and the adjacent sidewalks; 2) polygons of pedestrian pacified streets or shared surfaces; 3) larger flowerbeds considered as blocking elements within pedestrian areas; and 4) pedestrian crossings. It is

important to note that the latter is not an exhaustive layer since many crosswalks are missing or not up-to-date, therefore a thorough manual revision was carried out for this research.

Secondly, for modeling the adjacent urban fabrics outside the municipality of Barcelona (L'Hospitalet de Llobregat, Esplugues, Sant Adrià, and Badalona), the cartography was retrieved from the Barcelona Metropolitan Area (AMB) geoportal, a repository that offers an updated line-based geometry of the whole metropolitan area excluding the city of Barcelona. For these areas, a manual transformation of the geometry into polygons and a manual redrawing of the geometry of the sidewalks were necessary as an intermediate step.

Finally, an additional layer of city blocks, buildings, or inaccessible areas was produced manually, starting from the geometries of buildings from the aforementioned databases and following a case-by-case manual assessment of the results. This lengthy process is essential to provide an accurate description of the free-accessible pedestrian areas during the day, which often includes private spaces, inner gardens, or collective spaces.

## 2.2. Modeling of the network

The workflow developed during the research is divided into three main stages (Fig 2): a) geometry processing to obtain the network geometry from the walkable polygon of the city, b) network definition for the conversion of the raw line geometry into a weighted directed graph, and c) quantitative analysis of the results and network centrality metrics.

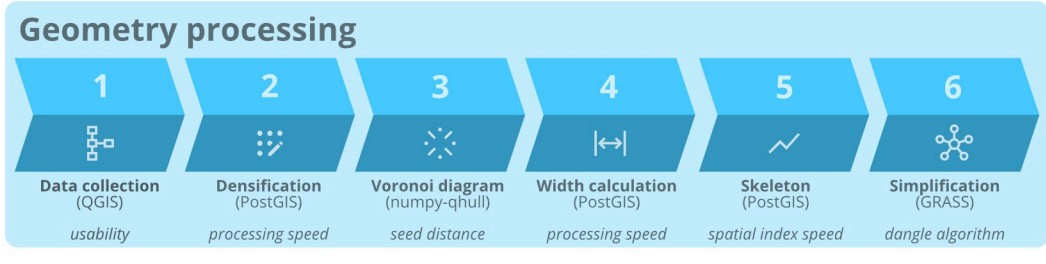

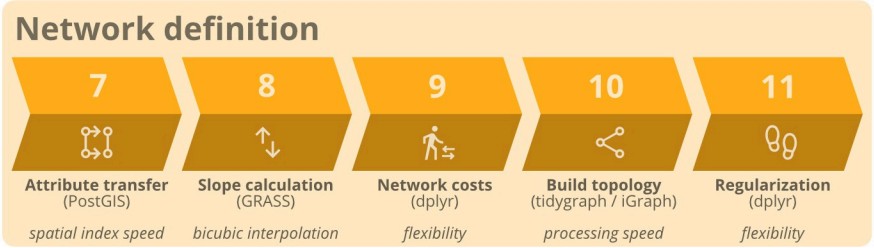

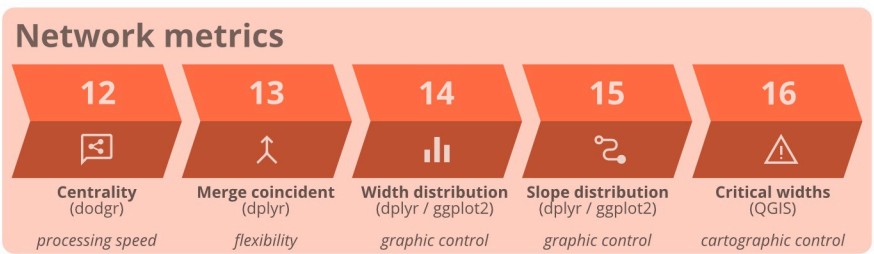

**Fig 2. Diagram of the workflow, with the three main processing blocks in separate rows.** Steps are numbered in the order they were performed, and the processing name appears in bold with the selected processing tool in parentheses below. The main motivation of the tool choice among alternatives appears in italics underneath. Source: the authors.

To make the research reproducible, the software and libraries used during the development were all open source, and the languages and IDEs used are R (RStudio), Python (Spyder), and SQL (pgAdmin). The main packages in the R language were the *tidyverse* metapackage (*dplyr*, *ggplot2*, *purrr*), *sf*, *sfnetworks*, *tidygraph*, and *dodgr*, and the libraries in Python were *pandas*, *numpy*, and *scipy*. The central database was PostgreSQL with the PostGIS extension, and the GIS tools were QGIS and GRASS.

## 3. Geometry processing

### 3.1. The geometry of the sidewalks polygon

During the first phase of the process, the geometry of the polygons of sidewalks, pedestrian areas, and crosswalks in the study area were produced. It should be noted that the polygon construction process was automatic but involved the manual review of pedestrian crossings, the delineation of 'inaccessible' and 'accessible' areas such as interior patios, stairsteps connecting different levels, street-level passages through buildings, or abrupt changes in topographical levels. For this fieldwork mapping, those private areas or those with restricted access control were ignored, so that the result offered an accurate representation of the sidewalks accessible during the day without restrictions.

The cartographic base takes into consideration large flowerbeds, such as those present on Diagonal Avenue, Sant Antoni Superblock, or Gran Via Avenue, but does not include smaller tree pits and other street furniture elements. Likewise, in large open areas, only paved areas or areas with easy pedestrian access were selected, excluding large green areas or any small constructions that made passage difficult.

The geometry was organized into three polygon layers: 1) the sidewalks platform edges (including the built parts of the blocks), 2) the city blocks (buildings or inaccessible areas), and 3) the pedestrian crossing polygons. Working with separate layers simplified the production of the cartography using CAD and GIS tools, as well as updating and correcting small errors in the geometries during the production process. However, this format was unsuitable for the following steps and had to be processed using standard GIS tools to produce a single entity that represented the contiguous walkable surface within the area of study.

The sidewalks and the pedestrian crossings polygons were dissolved (unary union) into a single layer, and the city blocks' geometry was subtracted from the unioned polygon in order to generate a single layer without any overlapping geometry. As this process produces some degenerate geometries, it was necessary to check and fix any topological errors resulting in invalid geometries, generally ring self-intersections or polygons with zero area.

With the fixed geometry, the parts of the resulting multi-part polygon were separated, and only the part with the largest area was kept, discarding unreachable islands. This result was visually inspected because the extraction of the largest polygon often revealed disconnects in the walkable surface that might require to be fixed in the case that they did not reflect the physical reality. Therefore, the process described was carried out in a feedback loop to ensure adequate quality control of the produced geometry, before proceeding with the next steps.

### 3.2. Geometry transformation

The resulting polygon represented the inner area of the walkable surface in the area of study without unreachable islands, and its limits consisted of a single (by definition) outer ring, and 28,845 interior rings. This polygon representation was not suitable for the analysis of the walkable infrastructure, and it was necessary the conversion to another simpler geometric representation, on which the analysis could be conducted.

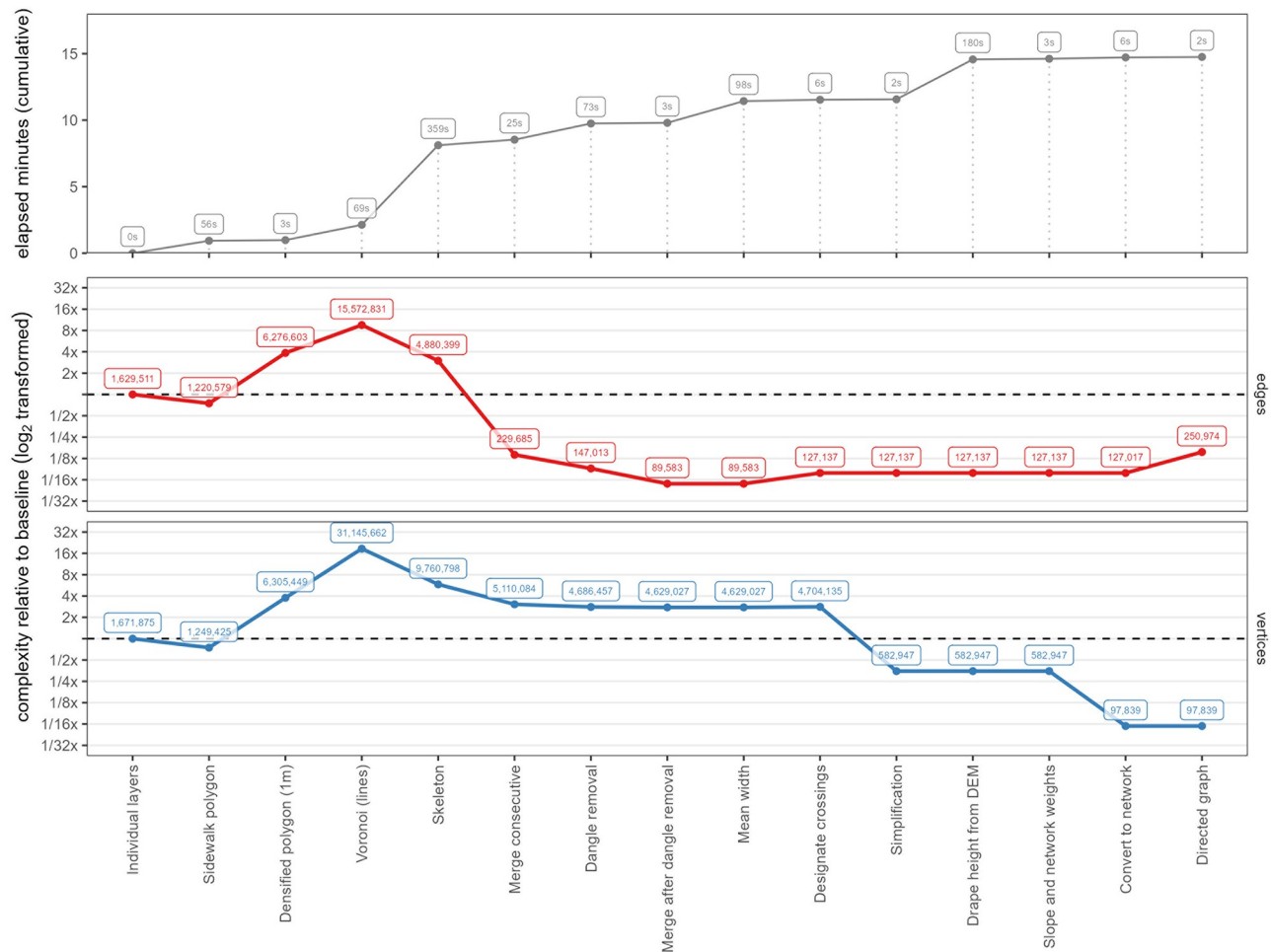

**Fig 3. Workflow steps to produce a directed graph from the source data.** The complexity of the data is expressed as node and vertex count at each step, in a logarithmic scale. The processing time required is shown as elapsed minutes from the initial condition. Source: the authors.

The network analysis required reducing the walkable surface geometry to a set of connected line segments using the medial axis transformation [49]. The geometry was initially processed with SFCGAL—which provides support based on the Computational Geometry Algorithms Library (CGAL) for Simple Features (SF)—through PostGIS, using the *ST_ApproximateMedialAxis* and *ST_StraightSkeleton* functions. Because of the complexity of the geometry, neither of the functions was capable of processing the input and returning successfully. It was, therefore, necessary to approximate the medial axis of the shape by implementing a workflow from scratch: 1) generating the Voronoi tessellation, 2) computing the width of the segments, 3) extracting the Voronoi segments inside the polygon shape, and 4) simplifying the network (Fig 3).

## 3.3. Voronoi tessellation

The Voronoi tessellation partitions a space (in this case a two-dimensional plane) into regions from a set of points (seeds) so that any point within these regions has the same seed as its nearest one. The boundaries of the Voronoi tessellation (ridges) define the limit where the nearest feature changes to a different seed, and therefore the points in these ridges are equidistant to

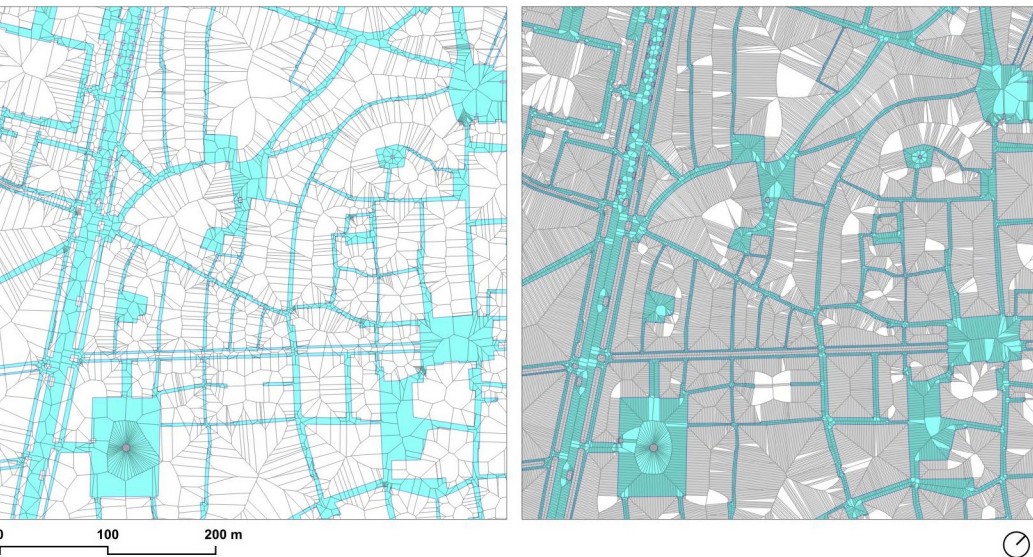

**Fig 4. Effect of the densification in the accuracy of the centerlines: Default sidewalk geometry (*left*) and densified sidewalk geometry (*right*).** Source: the authors. Base Cartography from CartoBCN, under CC BY 4.0.

two or more seeds. This property is useful in approximating the medial axis of the walkable polygon because to determine the centerline of a sidewalk we are interested in finding a sequence of points that are equidistant to the rings that define the boundaries of the polygon.

Therefore, the first step in constructing the medial axis was computing the Voronoi tessellation, designating the points in the rings that defined the limits of the polygon as seeds. To avoid inaccuracies in the computation of the tessellation, because the vertices in the boundaries of the polygon were not evenly distributed across the length of the borders [50], the plos geometry was densified using the *ST_Segmentize* function in PostGIS, using a maximum segment length of 1 meter. This densification increased the vertex count by around five times, with the segment length acting as a parameter that controlled the accuracy of the final result (Fig 4).

Because of the number of points and their proximity, the calculation time, memory efficiency, and floating-point accuracy were critical. Considering that only the boundaries of the regions were required but not the polygons defining the regions themselves, several alternatives were considered, including the PostGIS function *ST_VoronoiLines*—based on the Geometry Engine Open Source (GEOS)–, and the GRASS GIS *v.voronoi* module. The tool of choice was the *qhull* library [51], using the *scipy.spatial.Voronoi* function in the Python library *Spatial Algorithms and Data Structures* (*scipy.spatial*) as an interface.

In contrast with the other evaluated tools, the *scipy.spatial.Voronoi* function returns an object with multiple attributes that simplify the calculation of the distance from the medial axis to the boundaries of the polygon from which it was derived, because include the information of the seed points that originate each of the Voronoi ridges, allowing to accurately calculate the distance to the boundary of the original polygon efficiently (Fig 5), without requiring a spatial index, as discussed in the next section.

After running the calculation on the densified points of the polygon boundary, the function returned the data corresponding to more than 15 million tessellation ridges, and ridges going to infinity were automatically trimmed by the bounding box of the original points. The *qhull* library was able to compute the Voronoi tessellation in seconds on a midrange computer.

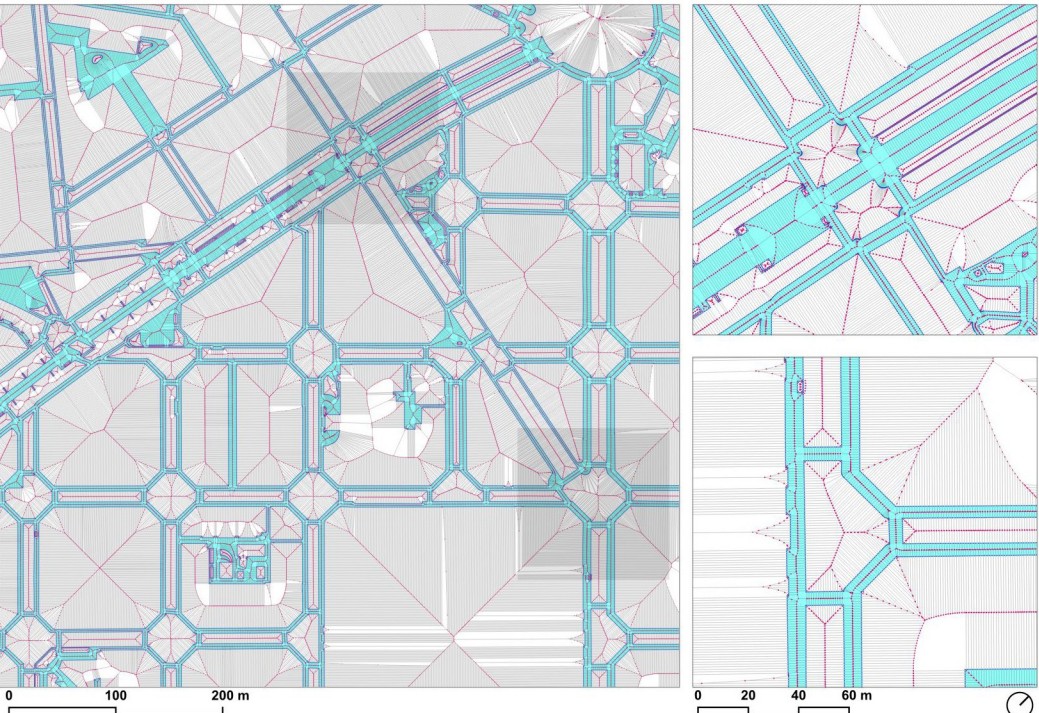

**Fig 5. Densified sidewalk polygons (*light blue*) with their originating seed points (dark blue), and resulting ridges (*black lines*) with their ridge end vertices (*red*).** Source: the authors. Base Cartography from CartoBCN, under CC BY 4.0.

## 3.4. Width calculation

The determination of the most representative sidewalk width [24] is challenging, especially for large sidewalk polygons with very irregular and complex shapes that contain numerous inner rings, as is the case of the typical walkable surface of a whole city when taking into account the presence of pedestrian crossings. In the case of this research, one of the objectives was to determine the width of all sidewalk segments corresponding to the skeleton of the polygon.

The brute force approach to determining the distance to the polygon edges was using the distance functions in a spatial database. However, this approach does not scale well because the polygon consists of a single entity, and therefore the operation cannot take advantage of any spatial indexing. Another more optimal approach was reducing the polygon to a set of rings (a single outer ring and collection of inner rings), where in this case it is possible to index the inner rings using their bounding boxes, but the issue of checking the distances against the outer ring remains unsolved because it consists in a single entity that by definition encompasses all the area of study. A third option would be reducing the perimeter of the rings to their vertices, essentially converting the problem to a point-to-point minimum distance calculation, which can be trivially indexed and partitioned to be distributed on all the cores of the processor. However, this approach would not be able to discriminate distances to points across the empty spaces in the polygon and the results would not be correct, because the distances have to be measured inside the sidewalks and not across the streets.

Therefore, because it was required to check the distance of a large number of segments (over 15 million) against an also large number of points (over 6 million) without the assistance of a spatial index to accelerate the calculations, a different approach was required.

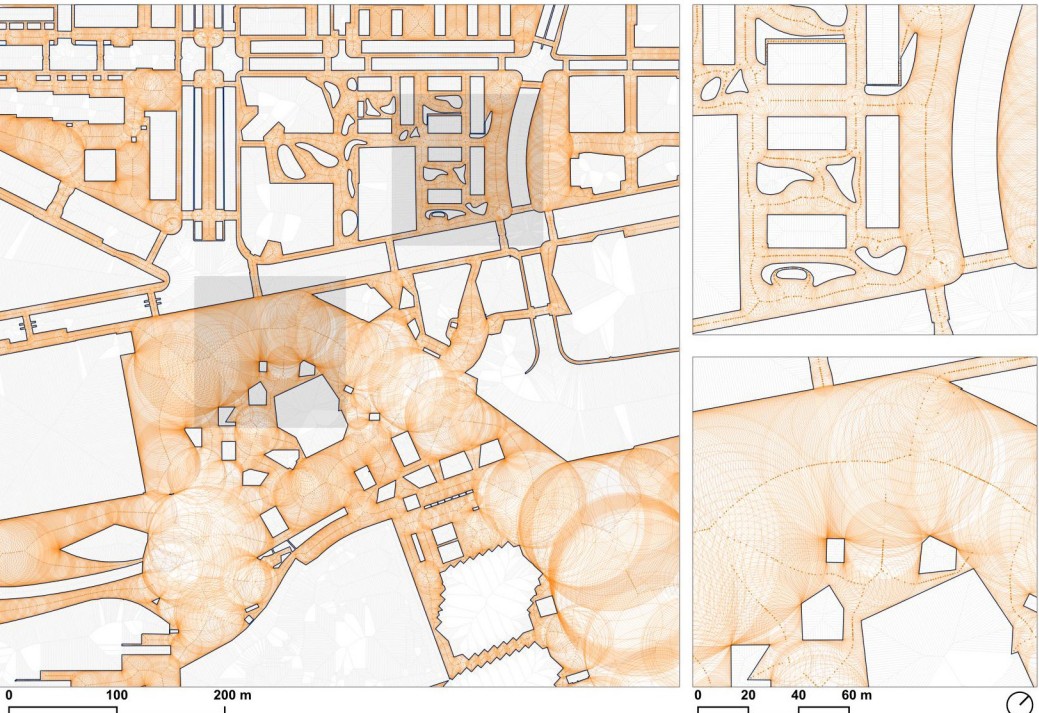

**Fig 6. Calculated widths of the extracted straight skeleton, represented as a circle with the computed width as its radius, with its center in the middle of the ridge segment.** Source: the authors. Base Cartography from CartoBCN, under CC BY 4.0.

In addition to these technical issues, estimating the width of a street segment requires two distance calculations, corresponding to either side of the axis if we consider that a line segment partitions the plane into two halves. Therefore, it was not feasible to simply conduct the minimum distance calculation twice, first to find the distance to the closest border, and then re-running the same computationally expensive calculation but excluding the element found on the first run, because both nearest points could be lying on the same side of the ridge segment.

To address this issue, the information of which pair of points originated each ridge segment resulted invaluable because it reduced the problem to two distance calculations to two already known point coordinates, which by definition were 1) the nearest points, and 2) at either side of the ridge segment; because of these optimizations, it was possible to compute the estimated widths of all line segments on a midrange computer in seconds as the sum of the distances to each seed point (Fig 6).

### 3.5. Straight skeleton extraction

The approximation of the straight skeleton was defined as the subset of the Voronoi ridge segments that were completely within the sidewalk polygon. However, checking this condition was not practical because of the large number of segments that had to be analyzed for inclusion within a single complex polygon that contained a large number of holes. In this case, it was possible to build a spatial index on the segments, but did not provide any advantage when checking a spatial predicate against a single entity that essentially covered the whole extent of the set.

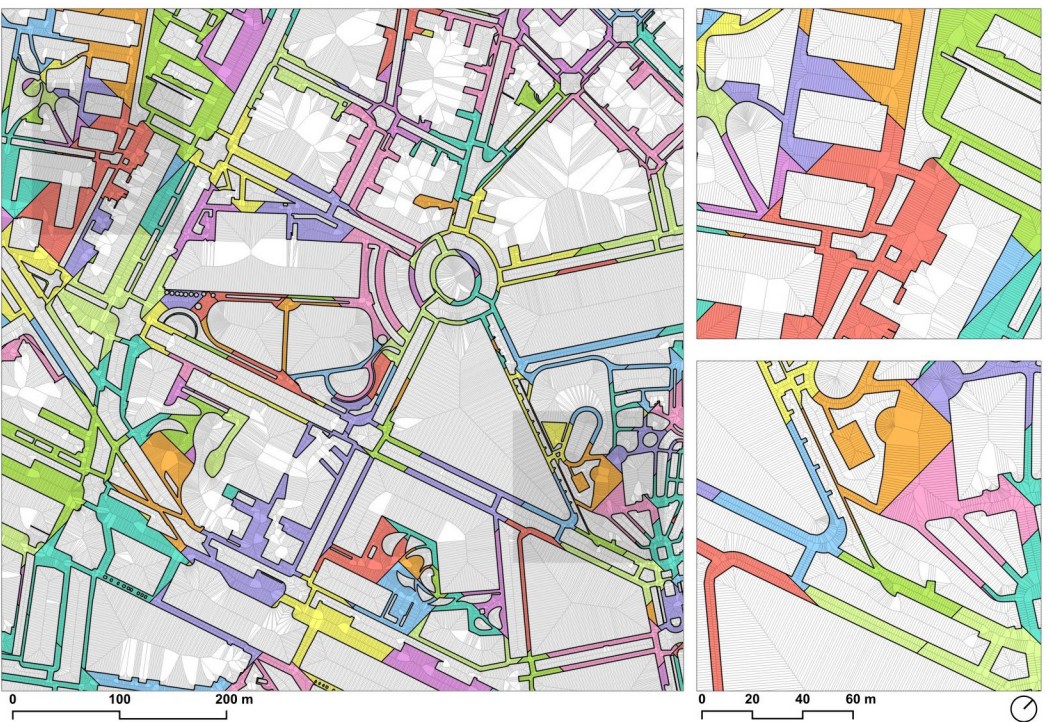

**Fig 7. Decomposition of the single sidewalk polygon into fragments enabling index-assisted spatial queries.** Fragments are randomly colored according to the modulo-10 of their feature identifier for differentiation purposes. Source: the authors.

The solution to the requirement of a spatial index of both sets of geometries was to partition the polygon into fragments, using the *ST_Subdivide* (https://postgis.net/docs/ST_Subdivide.html) function in PostGIS. With the default maximum of 256 vertices per polygon, the walkable area was decomposed into tens of thousands of fragments (Fig 7). These fragments could be indexed using the Generic Index Structure (GIST) in PostgreSQL, which produces an efficient R-Tree spatial index structure [52].

The resulting geometries (around 5 million ridge segments) constituted the approximate medial axis derived from the densified geometries at the intended 1-meter resolution, effectively deriving a network representation suitable for analysis from the walkable area polygon of sidewalks and crosswalks (Fig 8).

### 3.6. Geometry simplification

The resulting skeleton after keeping only the segments completely within the sidewalk polygon consisted of around 5 million line segments, with their corresponding width attribute. By construction, because they were derived from the Voronoi tessellation, the segments as former ridges of the diagram connected at their endpoints and therefore could be used in their present form to build a network topology. However, most of the nodes in this network had only degree 2, and therefore were not informative of the actual connectivity of the network, but were instead a side-effect of the densification step required to build the skeleton or the existence of segment approximations of curved geometries.

A second issue with the resulting network is that included some dead ends (*cul-de-sac*) resulting from small misalignments in the sidewalk geometry that were amplified during the

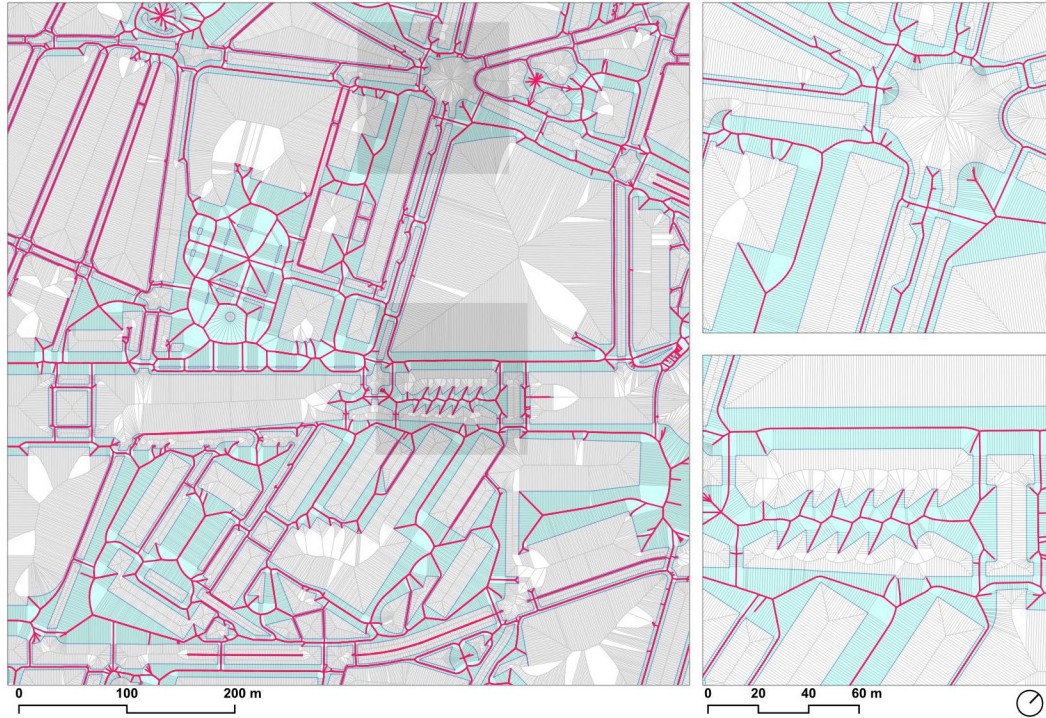

**Fig 8. Extracted straight skeleton corresponding to the ridge segments within the sidewalk polygon.** Source: the authors.

Voronoi tessellation, resulting in dangles in the network that were not representative of the actual connectivity of the sidewalk network, and therefore had to be removed.

The removal of this degree 2 nodes to simplify the network used a combination of the *ST_LineMerge* function in PostGIS and the *v.clean* in GRASS GIS. First, the lines were merged into line chains, reducing the number of edges in the network. With the resulting simplified geometry, the *rmdangle* tool in *v.clean* was used to remove the dangles longer than 50m, corresponding to approximately the width of the broadest streets in Barcelona (Diagonal Avenue and Passeig de Gràcia). The dangle removal process further reduced the number of segments (Fig 9).

In the resulting network without the dangles, some of the geometries could be further simplified, because in many cases the removal of a dangle resulted in the creation of a new degree-2 node where the removed dangle segment originally connected to the network. Therefore, applying the *ST_LineMerge* function again reduced the line segment count further.

The resulting simplified geometry of the sidewalks and walkable areas—with less than 2% of the original number of edges in the skeleton–, was used to summarize the width and slope attributes of its original constituent edges, in addition to the designation of those segments being crosswalks, as discussed in the next section.

## 4. Network definition

### 4.1. Width attribute transfer

The width was transferred from the individual widths of the skeleton segments using the *ST_Covers* function in PostGIS to identify a skeleton segment as being a constituent member of a given simplified line segment.

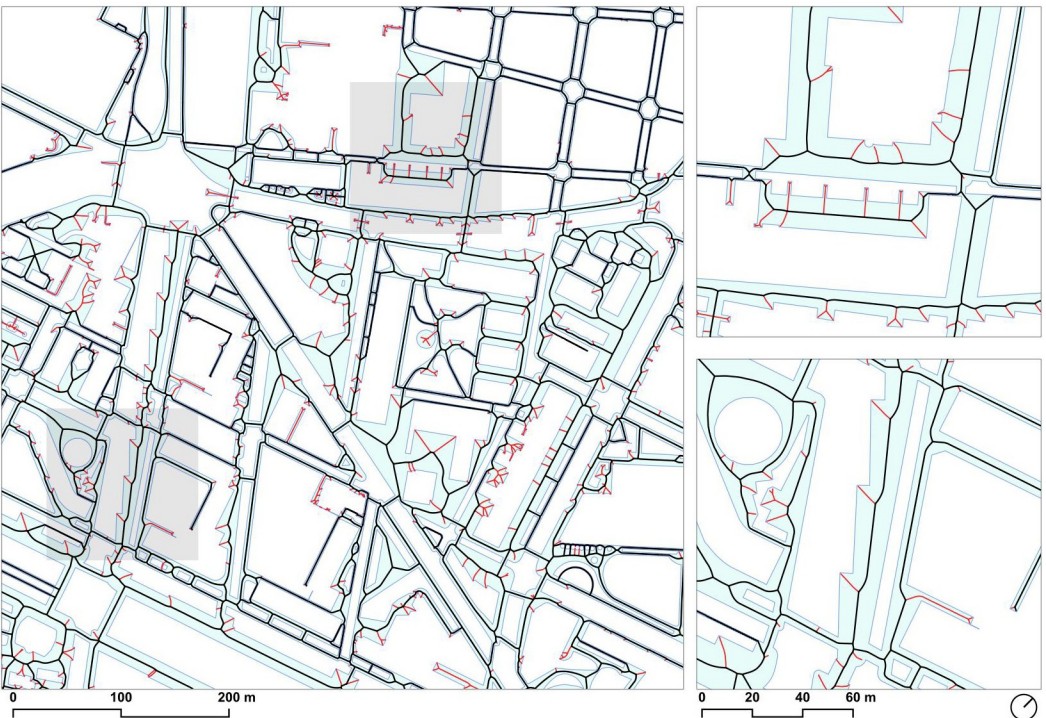

**Fig 9. Simplified skeleton in black, with removed segments identified as dangles in red.** Source: the authors.

For each of the simplified geometries, the widths of the covered skeleton segments were transferred as an attribute computing the weighted average of their widths, using the length of each skeleton segment as the weight.

## 4.2. Pedestrian crossing transfer

In order to transfer the information of crossings into the new skeleton, the geometries of the pedestrian crossings polygons (Fig 10) were used to split the geometry of the simplified network. The new geometry included a new Boolean attribute (*is_crossing*) that captured whether the segment was a pedestrian crossing or not. The special cases where a) a pedestrian crossing crossed multiple line strings, or b) a line string was crossed by multiple pedestrian crossings were considered. This required using both difference and intersection operations on both geometries.

## 4.3. Slope calculation

For the determination of the slope for each segment in the network, the 2-meter pixel resolution Digital Elevation Model (DEM) available from the Catalonian cartographic authority was used as a source of topographic elevation data. However, the slope calculation was challenging because it required considering multiple factors when computing the slope of the network from the data in the DEM raster surface.

First, the pixel-wise slope computed with common raster tools cannot be directly transferred to a segment overlapping the raster, because there are infinite tangent lines to the DEM surface at any point (all those contained in the tangent plane). It was therefore necessary to compute the slope for each individual line segment, transferring the DEM elevation to the

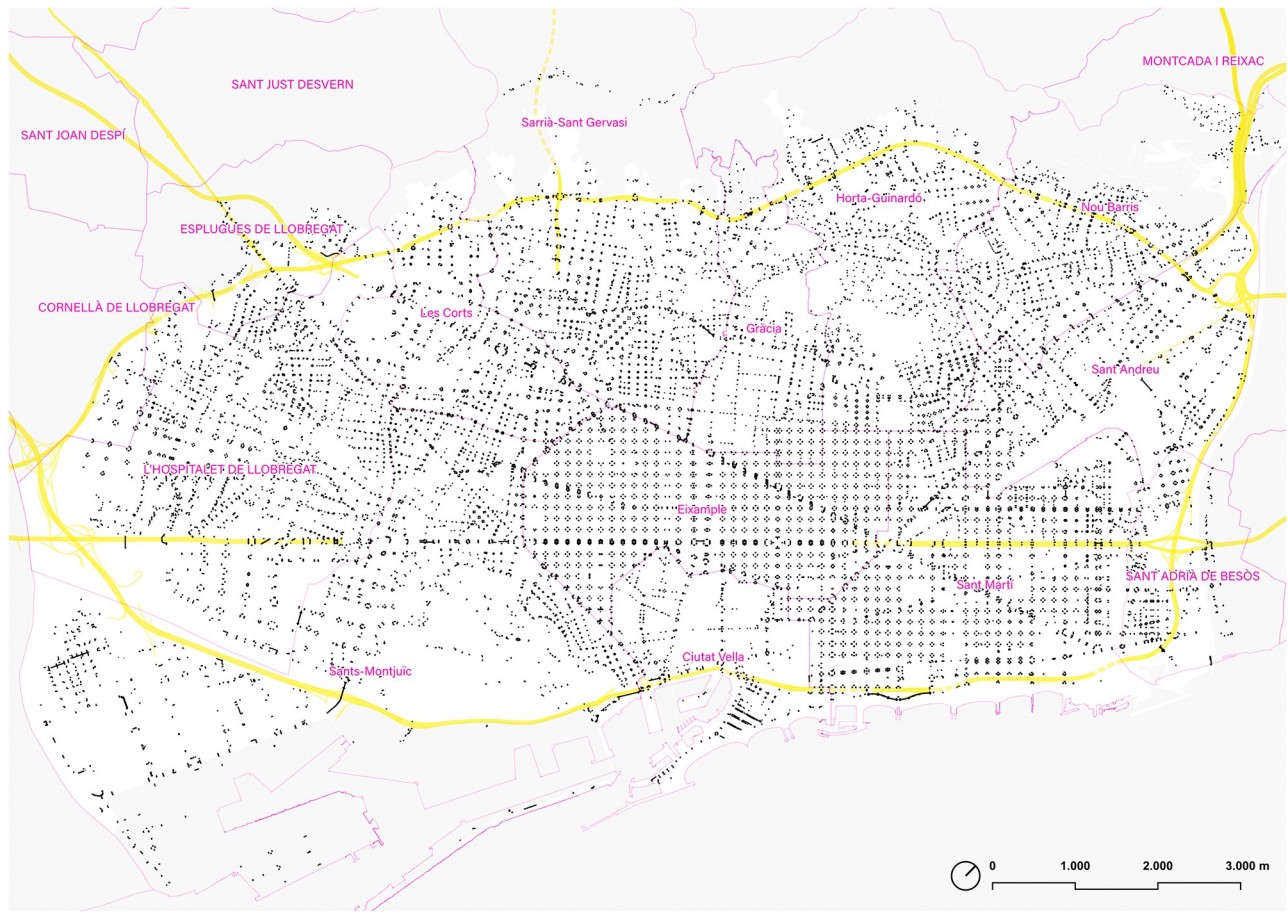

**Fig 10. Crosswalks of the 'walkable Barcelona'.** Source: the authors.

vertices of the network edges. This approach, while theoretically correct, introduced in practice a new source of error because of the interaction between the discrete nature of the DEM, and the separation of the nodes defining the geometry in the network edges, which could produce a very steep slope, an artifact not representative of the reality of a street segment.

To address these issues, the edges were first simplified in PostGIS to reduce the number of nodes defining its geometry using the Douglas-Peucker algorithm [53], followed by a densification step with a higher value to avoid producing very long segments that could not capture the actual topographic changes. The Z values from the DEM were transferred to the line segments using the *v.drape* (https://grass.osgeo.org/grass82/manuals/v.drape.html) command in GRASS GIS. The Z was interpolated using bicubic interpolation because the function *ST_SetZ* (https://postgis.net/docs/RT_ST_SetZ.html) in PostGIS only offered bilinear interpolation at the time of writing. The result was the DEM height attribute transferred to the nodes in the network edges, avoiding the pitfalls discussed above.

## 4.4. Network costs imputation

To accurately model the network costs from the slope and crosswalks information, and reflect their effects on walking mobility in a spatial network, the following properties had to be considered:

1. The slope of a segment is dependent on the direction across the segment because uphill and downhill interpretation is dependent on the forward or reverse direction

2. The effort (network cost) of the slope is not linearly proportional to the slope value, either measured in percent slope or as an angle (degrees or radians)

3. A network edge can consist of multiple uphill and downhill segments with different slopes, with the cost consisting of the sum of the costs of each individual segment

4. Crossing the street in a designated crosswalk represents a penalty in the speed of travel because pedestrians have to stop or slow down

To account for these properties, the cost calculations had to be conducted for all segments that constituted each network edge, in either direction, to obtain a pair of network costs per edge, one in the forward direction and one in the backward direction. The walking velocity was adapted from the hiking formula provided by Tobler [54], but changing the original maximum hiking speed of 6 km/h (around 1.7 m/s) to a more conservative multiplier in urban settings of 0.9 m/s (Fig 11).

In the adapted formula of $W = 0.9 \cdot e^{(-3.5 \cdot abs(S+0.05))}$ where S is the slope (measured as rise/run), the maximum walking speed on flat terrain is 2.72 km/h instead of 5.04 km/h in Tobler's formulation.

The adapted formula was applied to all individual segments that constituted a network edge, obtaining the speeds derived from the slope calculation. However, as the speeds could not be used as network costs—because of the different lengths of the segments–, they were converted to times in seconds dividing each distance by the corresponding speed. The use of time as network cost also had the advantage that it could be integrated with waiting times in pedestrian crossings, with an estimated average penalty of 30 seconds per crossing, calculated according to daily experience in Barcelona. Future research will adjust the waiting time according to each type of crossing.

Finally, the network costs were summarized per network edge summing the individual costs of the segments in which they had been decomposed. Two costs were computed, one in the forward and one in the backward direction, for the following three different scenarios:

- Distance cost only (as a baseline)

- Time considering slope only

- Time considering both slope and pedestrian crossing penalty

## 4.5. Network topology construction

To conduct further network analysis, it was necessary to derive a network topology from these linestring geometries and obtain a network representation following the principles of tidy data [55]; in this representation, the network graph is represented as two related tables, one corresponding to node data and another corresponding to edge (connections between nodes) data. The *tidygraph* 1.2.2 [56] R package was used as the framework to support this network representation efficiently, taking advantage of the spatial data types support provided by the *sfnetworks* 0.6.1 [57, 58] R package, which integrates SF support through the *sf* 1.0–9 [59] R package as a geometry column in both the node and edge tables. The *cityseer* [60] Python package was also considered but did not offer the same level of integration with the other R software tools used during the research.

With the network stored as a *tidygraph/sfnetworks* object, the maximally connected components of the graph were identified using the *iGraph* 1.3.5 [61] R package. The component with

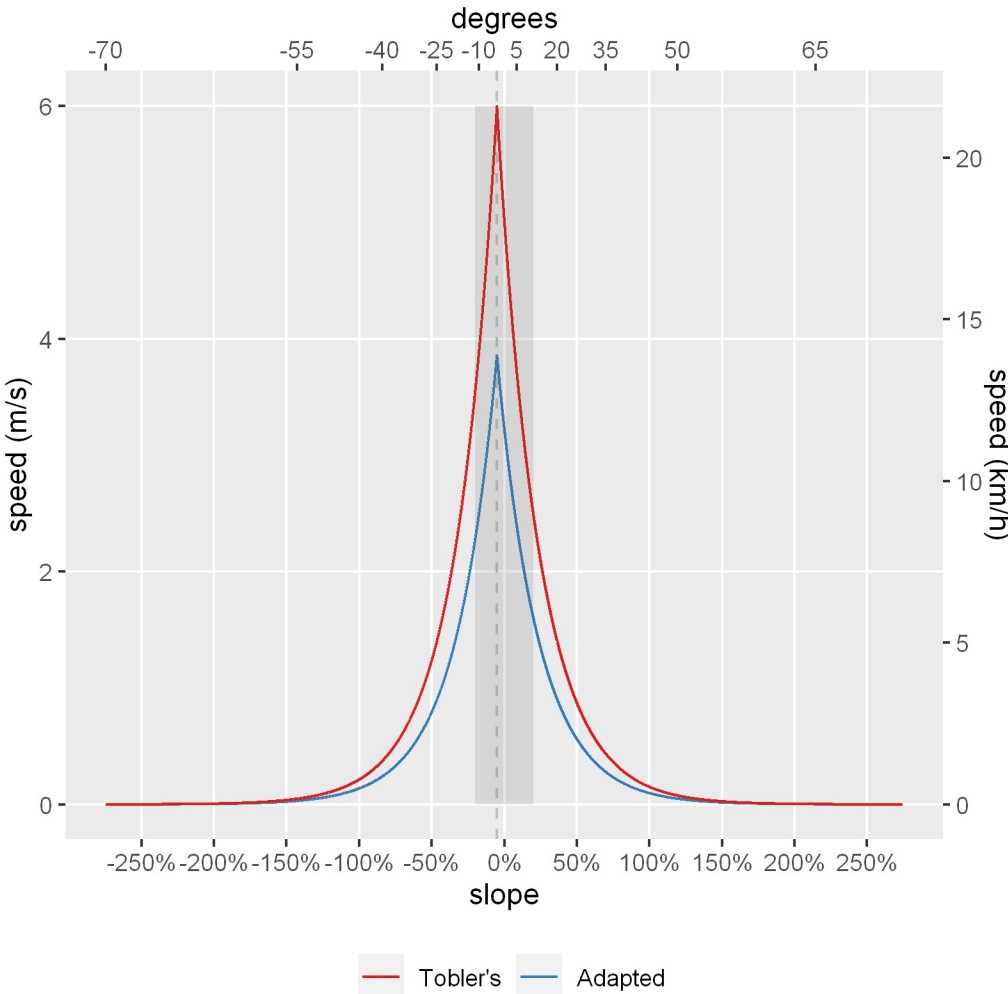

**Fig 11. Comparison of Tobler's original hiking formula compared to the adapted formula.** Maximum walking speed (5% downhill slope) denoted with a gray vertical dashed line. Walkable slope ranges (lower than 20%) are shaded in gray. Source: the authors.

the largest number of nodes (giant component) was identified and all nodes not belonging to this component were discarded along with any related edges, obtaining a graph without isolated subgraphs, i.e. unconnected sidewalks and therefore unreachable from the main network, which is a requirement in the majority of network centrality definitions because they assume a connected graph.

### 4.6. Compensation for the heterogeneity of node density

The betweenness centrality (both in its node betweenness and its edge betweenness variants), computes the shortest paths between all pairs of nodes and aggregates the number of times each node or edge is traversed. The shortest paths are defined as the ones passing through the smallest number of edges, or in the case of weighted graphs, the ones where the sum of its weights is minimized. However, in the case of spatial networks, the street representation of the urban fabric is oftentimes very heterogeneous; because all paths are evaluated between nodes,

this heterogeneity makes the calculation of spatial network centrality measures sensitive to the node density.

For example, in complex urban patterns such as the case of Barcelona, areas with large blocks have a relatively sparse distribution of nodes, while structures with smaller blocks tend to contain a higher spatial density of nodes, *ceteris paribus*. Furthermore, in addition to block size, the block shape can also introduce a bias in node density, as in the case of Manhattan, where edges along streets have a higher node density than avenues running perpendicularly. In addition, the actual geometric definition of the network can introduce nodes in specific areas, for example, to define the shape of a curve, yielding a spatial distribution of nodes that is not representative of the actual street network topology.

It was, therefore, necessary to compensate for this spatial network heterogeneity regarding the density of nodes to be able to accurately model the sidewalk network without introducing biases. The approach consisted in adjusting the weights according to the physical length of the edges. This transformation effectively cancels the effect of uneven node distribution, because the total sum of the edge lengths is evenly distributed in network space.

Since the focus of the study was the geometric attributes of the network regardless of their intensity of use, weighting according to the population was not considered adequate, as origins and destinations were not modeled. However, it was still necessary to perform homogenization for the reasons discussed above. In this case, each node was assigned a weight corresponding to half the length of all its incident edges (Fig 12). Since all edges had only two endpoints by definition, the weight was evenly distributed by the edge to each of the terminal nodes, and because the nodes aggregated the sum of these half-distances, the sum of weights was the total length of the network.

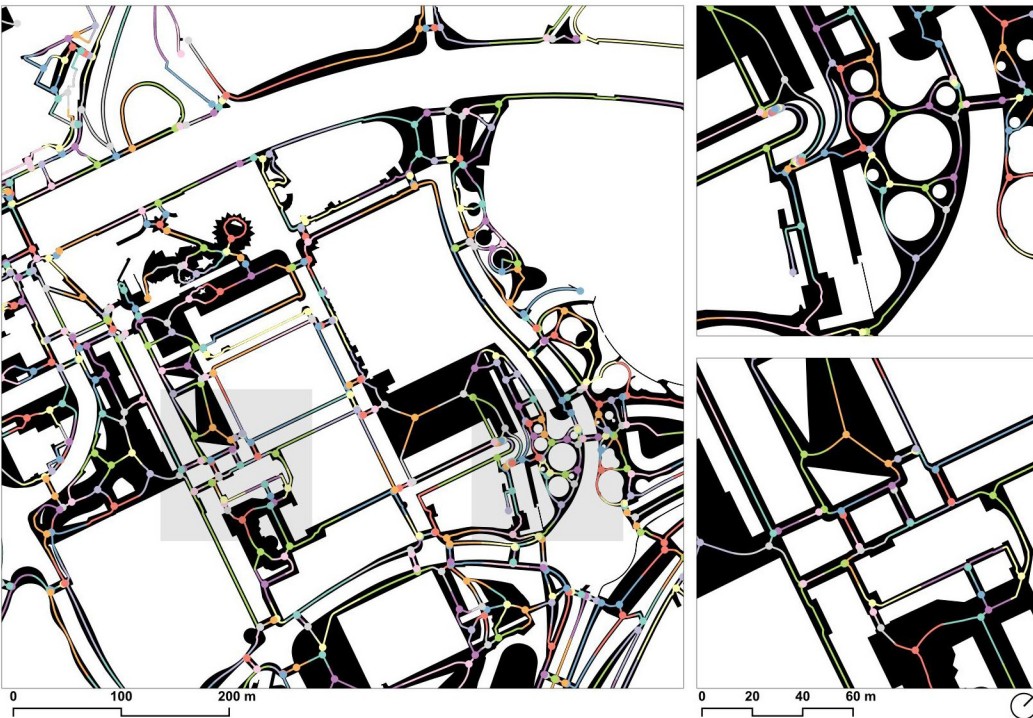

**Fig 12. Graphical representation of the homogenization process where each node receives a weight corresponding to the sum of the half-distances of all its incident edges.** Nodes and half-edges are randomly colored according to modulo-10 of the feature identifier of the terminal node for differentiation purposes. Source: the authors.

## 5. Results

### 5.1. Centrality measures

The edge betweenness centrality calculations were conducted using the *dodgr* 0.2.16 [62] R package, which provides efficient algorithms that implement the parallel computation of network routing in multicore machines. In addition to the calculation speed, it provides a critical feature over other engines, which is its capacity to support directed graphs with different weights in either direction. This feature was necessary to capture the contrast in walking uphill or downhill in the walkable network to accurately model real pedestrian behavior.

Because the package is still in development, it does not yet provide the functionality to convert from *sfnetworks* to *dodgr* objects, and it was necessary to write a custom routine for this conversion. The resulting directed graph in *dodgr* produced two logical edges for each physical edge, one for the forward direction and another for the backward direction (according to the ordering of the vertices in the linestring geometry). With the network in *dodgr* format, the redundant paths (about 1.2% of all edges) were discarded aggregating by common origin and destination pairs and selecting the smallest weight.

Three variants of the weights were considered in the directed graph: a) the network distance as a baseline, b) the time required to travel between origins and destinations considering the distance and the slope, and c) the travel times in situation 'b' but also considering the penalty to travel across a crosswalk.

The betweenness calculation was performed for the three different weights of the network graph, using three distance band thresholds for each of them, two using the equivalent of 15 minutes and 30 minutes on flat terrain without pedestrian crossings, and a third without limiting the distance walked. In a midrange computer, the first two calculations finished in about 5 minutes, while the unbounded distance calculation finished in around one hour.

### 5.2. Merge directed graph for analysis visualization

Because of the difficulty in representing simultaneously the forward and backward directions separately (because they were spatially coincident), both directions were merged back into a single physical edge from the two logical edges in the directed graph, to produce a new undirected graph where each edge had the sum of the values in each direction. This was only possible because the edge betweenness centrality result is itself by definition an aggregated count of the paths through an edge.

Because the edges were uniquely identified by the id of the origin and destination nodes, it was possible to join the edge geometry back to the results table using a table join operation, resulting in a linestring layer with these aggregated attributes, which were able to be visualized and analyzed further with standard GIS tools (Fig 13).

### 5.3. Quantitative analysis

The sidewalks and pedestrian areas in Barcelona constitute 22% of the total surface studied (1991 Ha). Surprisingly, this value is similar to the area dedicated to vehicle traffic on asphalt (21%, 1885 Ha). This similarity in the distribution of space coincides with the perception of the canonical street section of the Eixample grid designed by Ildefons Cerdà in 1859, where 50% of it was sidewalks and 50% for traffic. The rest of the surface of the studied area (78%) is dedicated to buildings, non-accessible enclosures, large flowerbeds, beaches, or green areas that are not accessible to daily pedestrian itineraries. On the other hand, the analysis of the medial axis gives a total linear distance of sidewalks and pedestrian paths of 3044 km. These reference values or their link to the number of inhabitants, portals, or activities could be a ratio

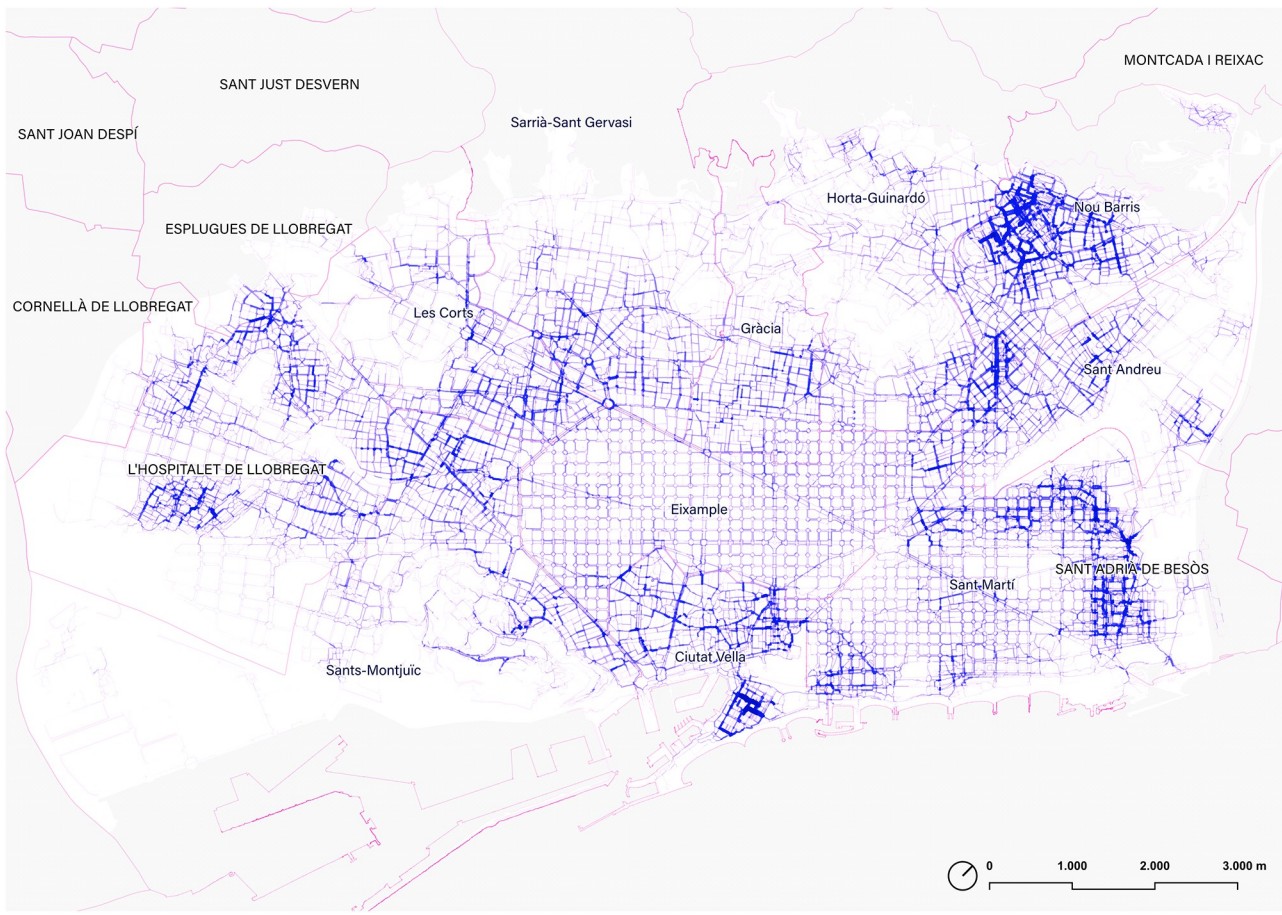

**Fig 13. Betweenness centrality at a 15-minute walking distance (0,9m/s mean speed, 810m equivalent distance), considering slope, crosswalks, and the topology of the walkable network.** Source: the authors.

of interest in the walkability of a city in comparison to other cities. In the studied area, for instance, the general ratio of sidewalks per person (according to Census Data 2022 provided by the City Councils of the studied municipalities) is 1.03 sqm/inhabitant. However, the sidewalks are not evenly distributed in the studied area, yet depend on the urban layout. Some areas have a similar surface of sidewalks but the ratio of sidewalks is different. This fact might point to the interpretation that sidewalks are used in different ways by locals, an argument that might be developed in future studies by crossing the geometric information with demographic data.

## 5.4. Sidewalk width distribution

The distribution of sidewalks by width is as follows: 14% of sidewalks are sidewalks between 0–2.4m; 24% are between 2.4–4.2m; 24% are sidewalks between 4.2-6m; 20% are sidewalks between 6-10m; 13% are sidewalks between 10-20m, and finally 6% are sidewalks over 20m wide. For the categorization of subgroups, the recent work carried out by the city of Milano [20] has been considered. Additionally, although the study for Milano defines the threshold of 'comfortable sidewalks' as 4.2m, in the analysis of Barcelona some additional subgroups have been added: from 4.2m to 6m—in order to identify the sidewalks of the Ensanche (5m)–; from

6m to 10; from 10m to 20m—considering the Ensanche total canonical street section–, and finally the sidewalks exceeding 20m wide.

The calculation of the width of the sidewalks gives a clear image of the variability of this value within Barcelona (Fig 14). Although the further urban interpretation of this result is required, some insights can be summarized. To start with, the first group of sidewalks (0–2.4 m) tends to highlight in red the centers of the historic towns which conforms to the large contemporary Barcelona conglomerate today. However, this distinction is not automatic, as cases in Gràcia or Ciutat Vella have narrow streets but the progressive pacification in the last third of the 20th century has 'widened' their sidewalks to the point of assimilating them to Ensanche fabrics (5+5m). Narrow sidewalks are a constant in neighborhoods as different as Pedralbes or El Carmel, the first marked by low-density and wide streets, and the second by uneven topography, higher density, and narrow streets. This map makes it possible to identify confrontational situations, as is the case of the streets of Barceloneta which today still accommodate sidewalks, parking, and vehicular traffic lanes in streets just 6.6m wide.

The map also shows the predominance of the sidewalks of the Eixample (around 5m) not only as a canon of the grid but also in some main avenues structuring the continuous Barcelona outside the Eixample grid. It could be said, therefore, that the sidewalk model of the Ensanche devised by Ildefons Cerdà is also a canon used in the city's avenues. The fact is that this visual consideration is also complemented by the fact that the weighted median value of

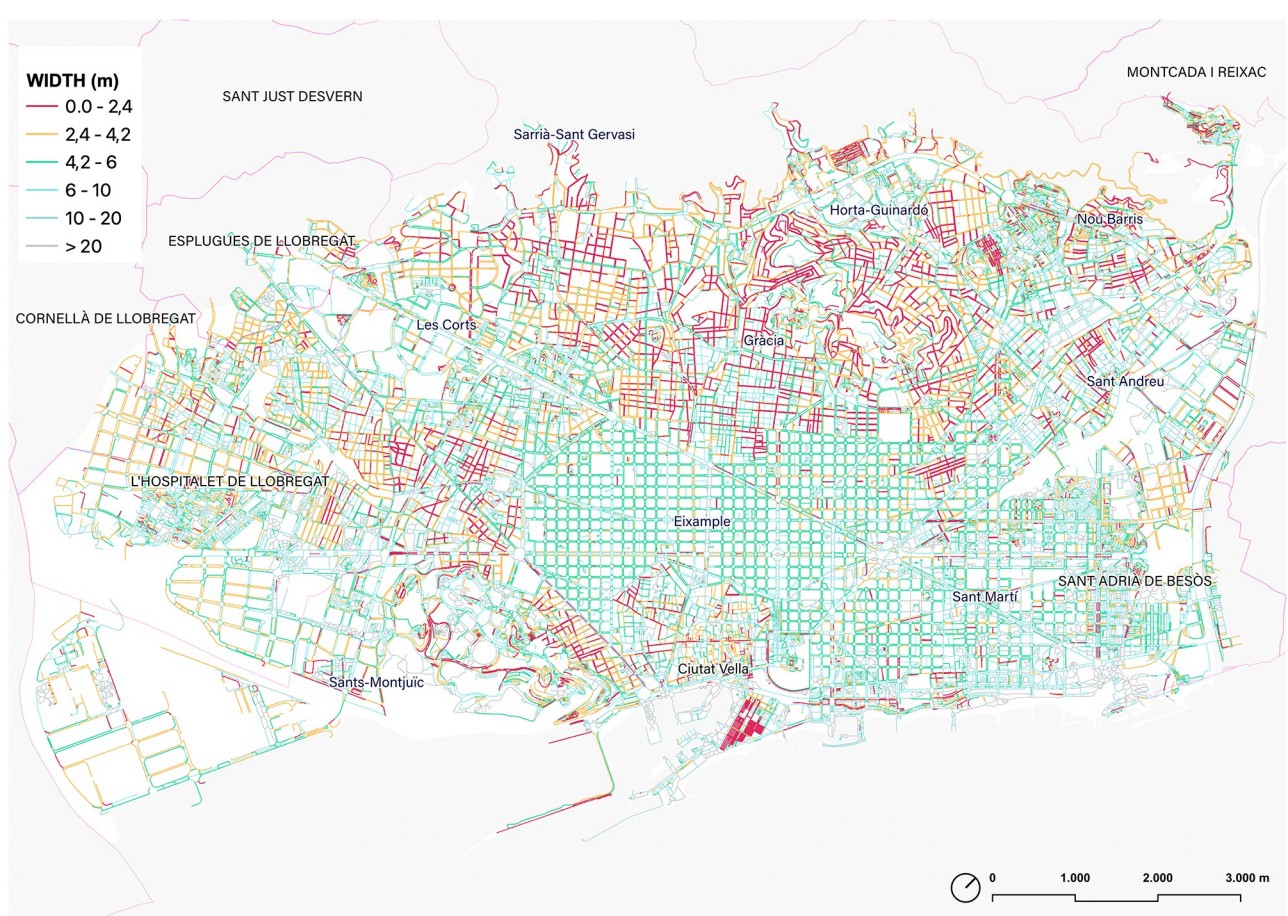

**Fig 14. Map of the sidewalks of Barcelona according to the width.** Source: the authors.

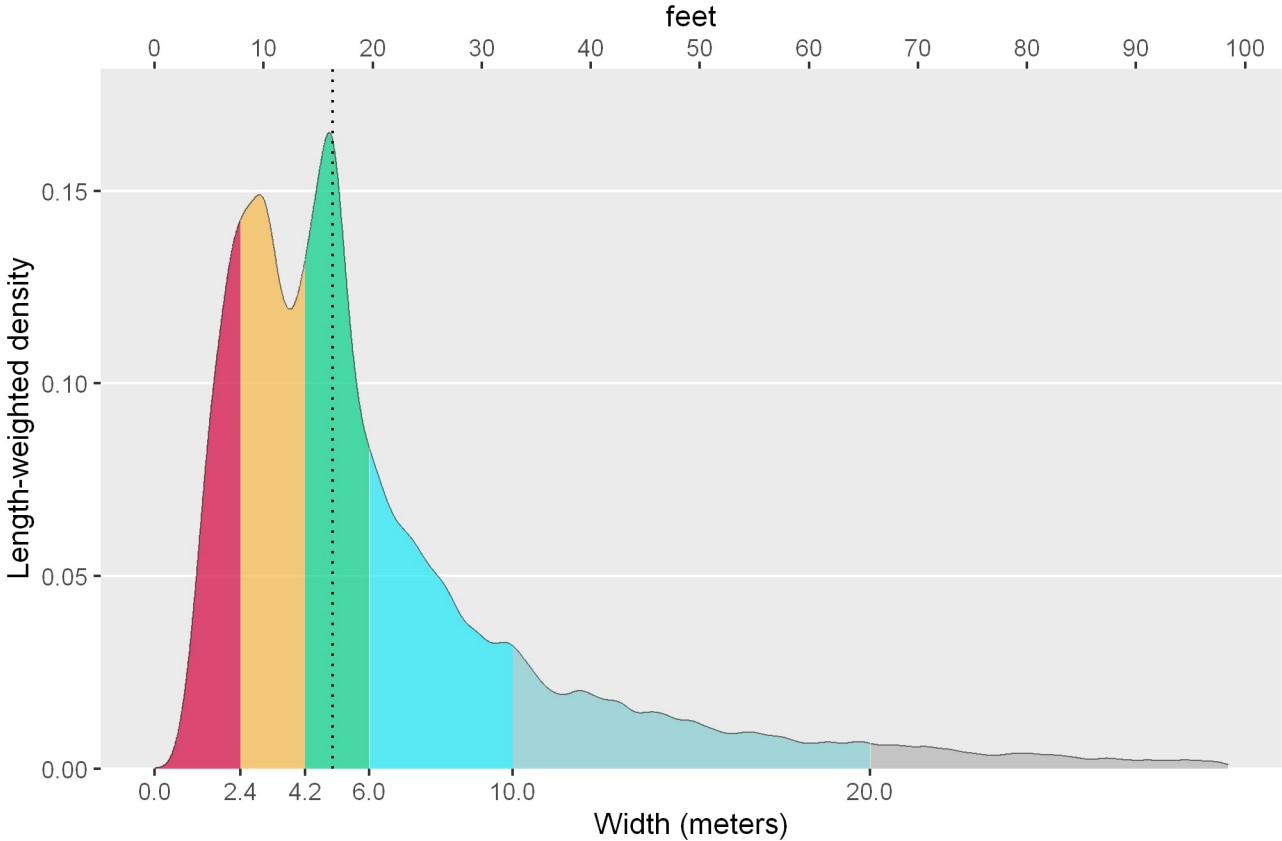

**Fig 15. Density distribution of the width of the extracted sidewalk segments, weighted by their length.** The vertical dotted line corresponds to the weighted median. Source: the authors.

width distribution of the 'walkable Barcelona' is exactly 4.98m, which clearly highlights the weight this sidewalk's width has in Barcelona (Fig 15).

The range of sidewalk widths between 6m and 20m (blue hues) is, in the first instance, an image of the higher street hierarchy of Barcelona, mainly identifying the major axes or structuring arteries. However, here it is possible also to use this range to highlight the 'boulevard' type section, whose central promenade does not usually exceed 20m. This is the case of the well-known Les Rambles (in the city center). Also, the figure is also very helpful to identify the open block urban typology [63], that is configured with large open spaces and where consequently the sidewalk space expands to form platforms, squares, and wide spaces.

Finally, the map shows the walkable platforms representing the sidewalks with more than 20 meters (gray). This figure is especially relevant to depict the prominent seaside promenades, the Fòrum platform, the area of the Anella Olímpica de Montjuïc, or the pedestrian areas that surround part of Sants Station. Also, some specific urban areas are also highlighted here, like Passeig de Gràcia lateral sidewalks (shared surface) or some areas in Poblenou connected to the development of superblocks.

### 5.5. Sidewalk slope distribution

The 'walkable' Barcelona is mainly defined by a wide plain gently sloping towards the sea and located between the deltas of the Llobregat and Besòs rivers. This plane is embedded between

the mountains of Montjuïc, the Collserola mountain range, and marked by the Tres Turons park and other less prominent hills. This distinction is evident in a quick survey of the general topography also represented in a map (Fig 16). However, when analyzing the effectiveness of the network of sidewalks, further attention to the subtle changes in slope is required, as this is a parameter that clearly marks the comfort of the walk and can influence the movement in the city [27, 28].

Without getting into extensive urban interpretations, which will be addressed in future research, some remarks might be useful at this point. Firstly, the slopes under 2% (in blue) represent 47% of all the sidewalks, and this slope is considered a flat surface according to the walking experience and urban design practice guidelines. These sidewalks mark the longitudinal direction of the Eixample (Besòs-Llobregat direction), a guideline that is very relevant in the daily experience both on foot and by bicycle. These itineraries are much more comfortable than transverse movements. On the other hand, the presence of flat areas in the city also makes evidence of the aforementioned deltas now occupied by Bellvitge in L'Hospitalet de Llobregat, Zona Franca, or Sant Martí.

The map also identifies the sidewalks with the greatest accumulated slope in each section, i.e. slopes greater than 4% (from light green to red, 31% of the total length of sidewalks) as shown in the slope density distribution (Fig 17). The result reflects not only the distribution of the general orographic shape of Barcelona, but it is also useful to evidence the subtle changes

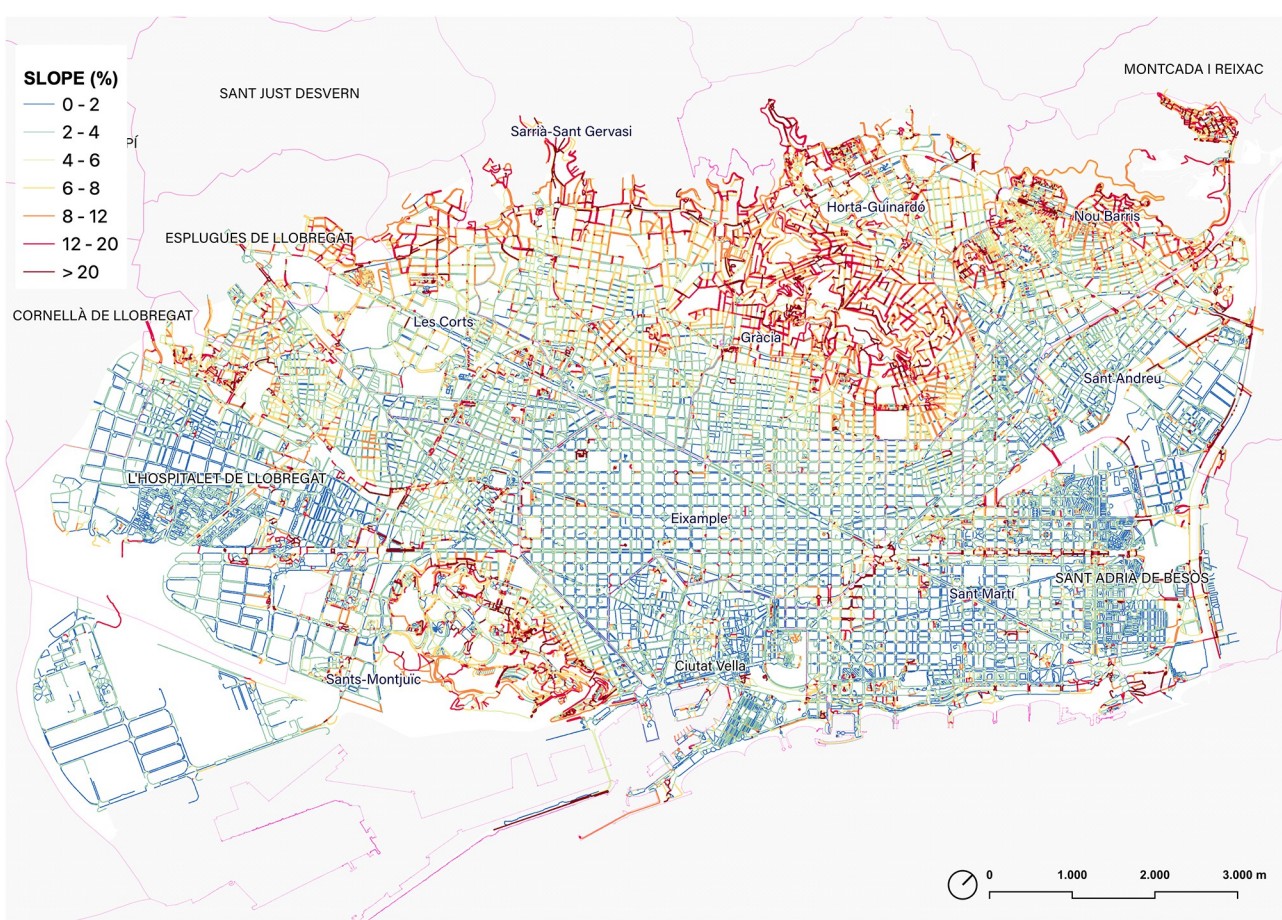

**Fig 16. Map of the sidewalks of Barcelona according to the slope.** Source: the authors.

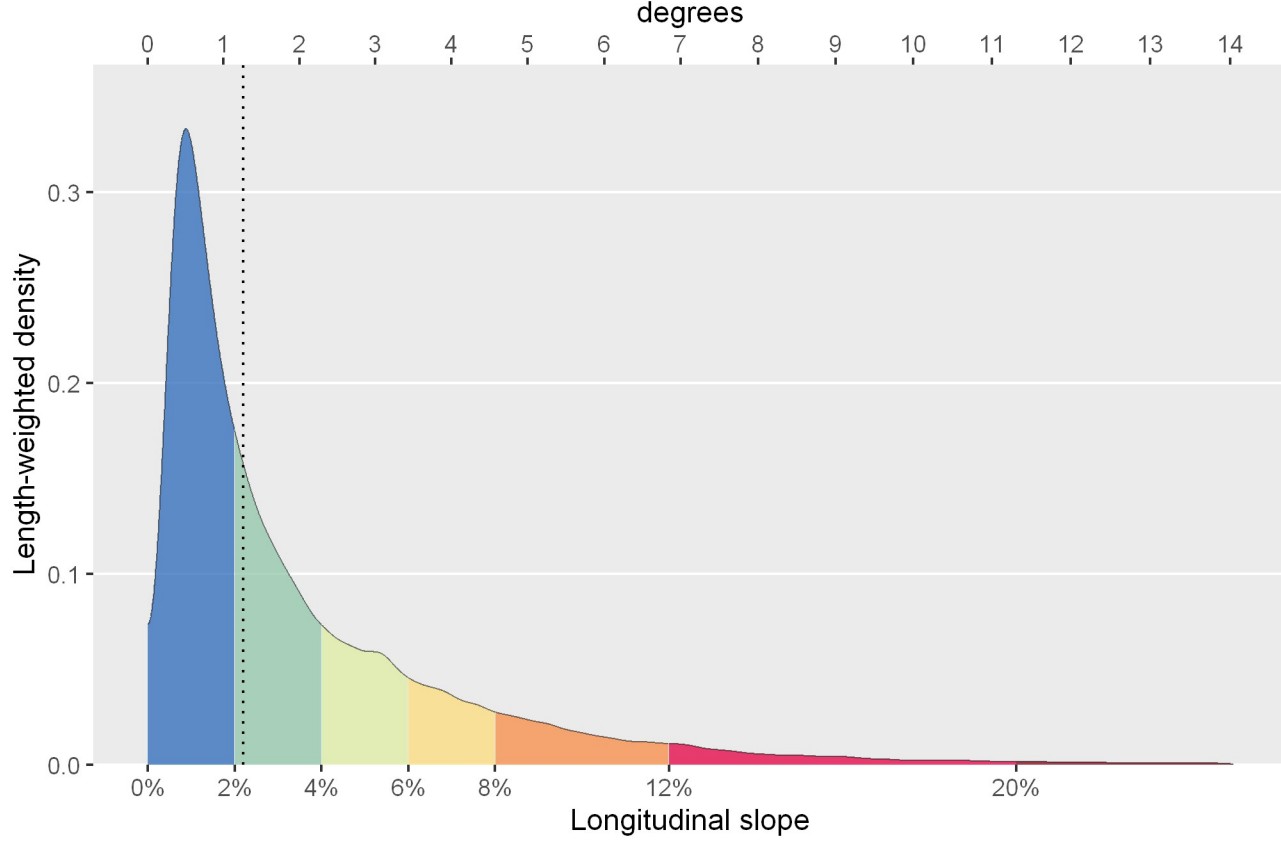

**Fig 17. Density distribution of the longitudinal slope of the extracted sidewalk segments, weighted by their length.** The vertical dotted line corresponds to the weighted median. Source: the authors.

in slope in flat areas, those micro-discontinuities which clearly mark the difference in urban continuity perception. In short, this workflow allows a clear recognition of those potential discontinuities that qualify pedestrian continuity in the 'walkable Barcelona'.

## 6. Conclusions

The method described in this research has been capable of generating a high-resolution network of sidewalks and pedestrian spaces. As the complexity and size of the sidewalks polygon and its transformation into a single skeleton require a large number of calculations, the library of choice was *dodgr*, because its algorithms were, at the time of writing, more efficient than the ones offered by *iGraph*, especially when computing centrality measures using multiple cores on directed graphs. At the same time, although this method has been applied to the analysis of Barcelona, the richness of situations of this case study might be enough to confirm the availability of this method to compute other cities with sidewalks polygons, pedestrian crosswalks differentiation (if needed) and a DEM topography model in order to take into consideration the orography.

Secondly, it is important to note that the development of this workflow has been developed using existing raw data and cartography from Barcelona and adjacent urban fabrics without taking into consideration street elements such as small flower beds, benches, or other urban objects which might mark a difference in the calculation of the width. However, if this data

was available, the process discussed would be able to process geometries with a much higher degree of complexity, as only 60 minutes are needed to perform the geometric calculations and measure the centrality of the network for walksheds under 15 minutes. Ongoing research by the authors is already exploring this potential in order to provide more detailed results.

Additionally, although the modeling of sidewalks is based on a partially manual review of the walkable polygon analyzed on September 1, 2022, the subsequent geometric processing and network analysis is designed to be done automatically. This is especially crucial for the sidewalks, an urban element that is constantly changing and redesigned according to people's needs and political agendas. Another interesting aspect is examining those streets which are pedestrianized at weekends, following the local initiative called 'Obrim carrers' (https://www.barcelona.cat/obrimcarrers/en Accessed 13 February 2023). This is a dynamic condition that could only be addressed by an agile calculation workflow of the pedestrian network such as the one presented in this research.

The betweenness centrality analysis has been based on the imputation of costs of the slope by translating it into a 'time' variable and using 3.24 km/h (0.9m/s) as the standard walking speed. This value might be easily adapted in order to analyze different social groups (kids, elderly people, etc.). On the other hand, it must be acknowledged that the calculation of costs of pedestrian crossings universally considering 30 seconds might not accurately describe the specific particulars of each situation. Some of the crossings are controlled by traffic lights with different waiting times, while others are only working as indicative areas for crossing without traffic lights due to the lack of constant and speedy traffic. The simplification used in this network analysis might be refined if data is provided but the current workflow is capable of designating the crossings as segments of the networks with a penalty. Finally, a special mention of the practice of jaywalking might be also taken into account, as it might change the essential conceptualization of a walkable network formed by parallel platforms on each side of the street and simply connected by crosswalks.

The research offers also an original adaptation of the betweenness centrality calculation to describe the urban walking experience, considering the crossings and the slope in order to evaluate each itinerary. For this work, an even distribution of origins and destinations has been used, not taking into consideration the role of existing urban attractions. This might be easily implemented as the *dodgr* library used in this workflow easily allows the linking of data to each node. However, the interpretation of the results is still significant and valid for explaining most of the movements of proximity since Barcelona presents substantially uniform compactness and density of inhabitants/Ha and, therefore, the sidewalks with a high level of centrality correspond to the most used according to daily experience. Although this argument will be validated by on-site measurements in future works, the fact is that the image offered in this paper gives an eloquent explanation of the potential of a given urban layout for walking.

The work is particularly applicable as it may allow the correlation between different metrics. If the relationship between *width* and *betweenness centrality* at an equivalent distance of 810m (15-minute walk) is taken into consideration, the result is particularly useful for detecting sidewalks with high potential of use and conflict or, alternatively, underused sidewalks by local movements. It must be remembered that the width *per se* is not directly connected with high urban quality, as wider sidewalks sometimes are not understood as human-scale itineraries. By correlating width and centrality, it is possible to identify those spots where the 'through movements' of people come into conflict with narrow sidewalks or *vice versa*.

These two extremes can be identified as two colors on a map (Fig 18). The most used sidewalks are highlighted in red and the least used in blue, always considering an even distribution of origins and destinations. At first glance, the map makes clear the conflict in many streets surrounding the Eixample grid. But it is also possible to find subtle differences within the grid

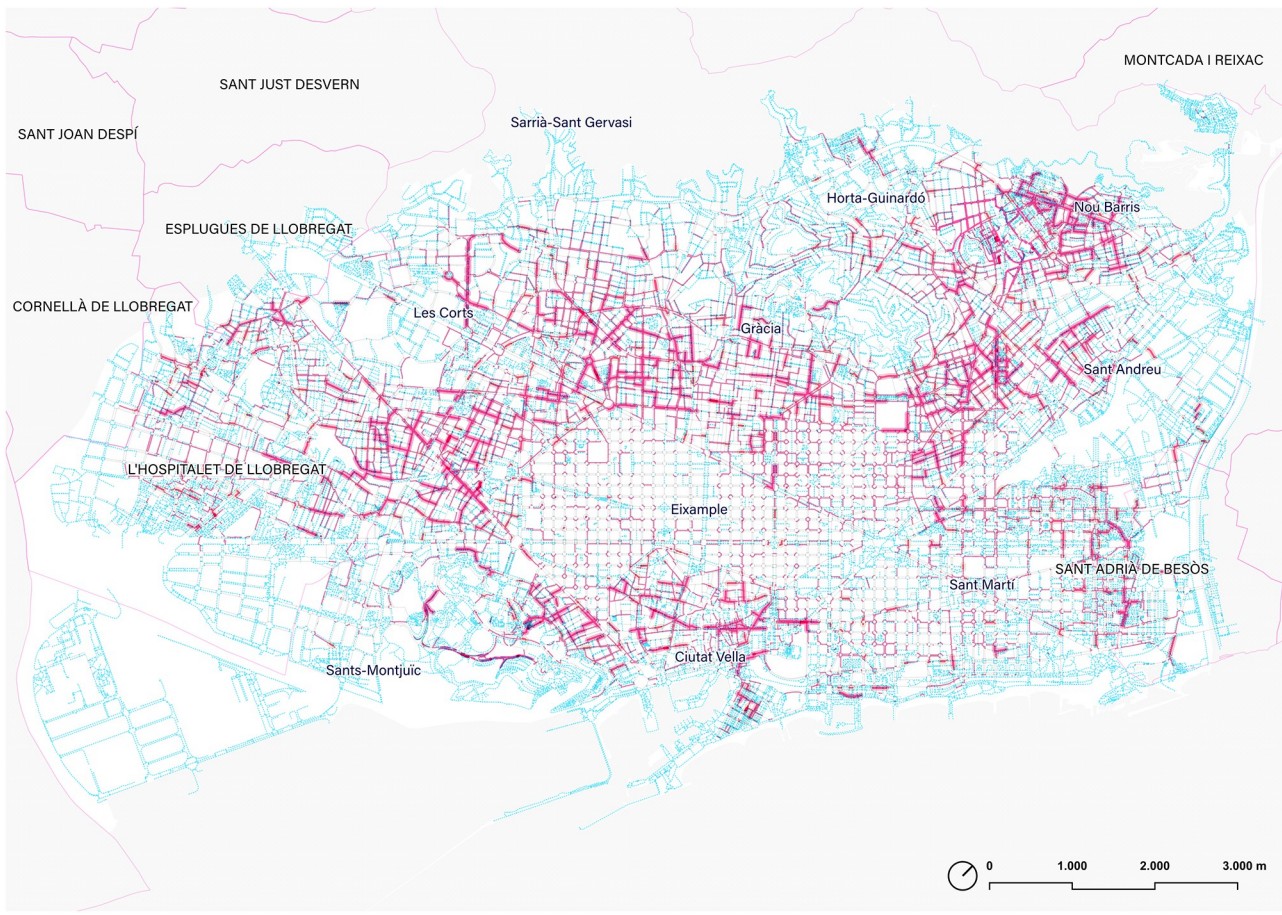

**Fig 18. Correlation map between betweenness centrality (15 minutes walking, 0.9m/s) and width of the sidewalks.** In red are those sidewalks with the highest coefficient, representing high levels of potential conflict between through-movement and sidewalks capacity. In blue, the walkable network with a low coefficient represents the less densified sidewalks by proximity movements. Source: the authors.

itself. In fact, as it is being explored in parallel research, the method is especially relevant to analyze the effectiveness of certain sidewalk configurations and, more specifically, to evaluate the impact of the octagonal chamfer corners of the Eixample in walking daily itineraries and other major irregularities such as diagonal shortcuts, passages, semi-open blocks, and public interior gardens. It is also being especially relevant for comparing the before-after configuration following the current green hubs and squares projects developed under the motto of the 'Superillas' (superblocks) and expected to be completed by 2023. The use of the method of this research might be of high interest to address evidence-based urban transformation processes by the municipalities.

The workflow is also helpful to provide evidence of the *continuity* of a network, as it is one of the most determining factors of any transport system and a key issue to achieve safe and efficient pedestrian itineraries. If walking continuity is addressed as a geometric problem detached from any environmental or experiential factors, it might be stated that it depends fundamentally on two factors: (1) the possibility of accessing any point in the walkable network without coming into conflict with another transport network; (2) the capacity of the network to use the most effective itineraries, taking into account minimum distances, whether metric, angular or topological. The first factor is fundamentally linked to crosswalks as points of

special conflict and impedance. The workflow can easily detect minor 'pedestrian areas' detached from the rest of the system. The second, on the other hand, is related to the detection of missing links in networks [64], i.e., strategic connections of the system that could clearly make a difference in the efficiency of the walkable network. Both aspects can be characterized by the workflow.

Thus, the method opens up a wide spectrum of future applications for urban studies. Firstly, the workflow is waiting for a more contextual interpretation of the result, now taking into account urban morphology, urban history, or social and economic data. But also, the research might be also of high interest to examine urban proximity models as the '15-minute city' [59], by giving an empirical discussion on the walkable platforms where this city is enhanced. In this sense, it should be emphasized that this first exploration of 'how Barcelona's sidewalks are' is aimed to shed light on 'how we use them' in future research. Here it will be necessary to include other factors that definitely influence the calculation of minimum itineraries as the minimum metric distance is not necessarily the most effective since it can be unpleasant to walk. Hence, the relationship between sidewalks and activities, demography, tourism, green areas, schools and facilities, or exchange nodes with other means of transport can be decisive for the modeling and evaluation of the quality of the walkable city, always on the basis of that no model will be able to fully explain the reasons *why* we walk but at least research should help to understand *where* we walk.

## Supporting information

**S1 File. Results of the calculation including the geometric parameters extracted (length and width in meters, average slope in percent rise/run, crossing designation as true/false), and the betweenness values for the combination of weights (distance, slope, or distance and slope) and distance thresholds in meters (810, 1620 and unbounded).** Source: the authors.
(CSV)

## Acknowledgments

The author would like to thank Celia Díaz and Belén Davila architecture students at the Barcelona School of Architecture (ETSAB) for their support in the production of the sidewalk polygon.

The authors would also like to thank the anonymous reviewers who, with their generous remarks, have substantially improved clarity in the presentation of the workflow, background, and results of the research.

## Author Contributions

**Conceptualization:** Francesc Valls, Álvaro Clua.

**Data curation:** Francesc Valls, Álvaro Clua.

**Formal analysis:** Francesc Valls, Álvaro Clua.

**Investigation:** Francesc Valls, Álvaro Clua.

**Methodology:** Francesc Valls, Álvaro Clua.

**Project administration:** Francesc Valls, Álvaro Clua.

**Resources:** Álvaro Clua.

**Software:** Francesc Valls.

**Supervision:** Francesc Valls, Álvaro Clua.

**Validation:** Francesc Valls, Álvaro Clua.

**Visualization:** Francesc Valls, Álvaro Clua.

**Writing – original draft:** Francesc Valls, Álvaro Clua.

**Writing – review & editing:** Francesc Valls, Álvaro Clua.

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
