## [Decision Letter · Decision Letter 0]

10 Feb 2023

PONE-D-22-35314Modeling Barcelona sidewalks: A high resolution urban scale assessment of the geometric attributes of the walkable networkPLOS ONE

Dear Dr. Clua,

Thank you for submitting your manuscript to PLOS ONE. After careful consideration, we feel that it has merit but does not fully meet PLOS ONE’s publication criteria as it currently stands. Therefore, we invite you to submit a revised version of the manuscript that addresses the points raised during the review process.

The manuscript must be corrected in all points indicated by the reviewers, such as:

1) Text is too long. If possible try to shorten it without losing quality;

2) In the webpages access date needs to be added;

3) There are very few scholarly articles cited in the text, and very many websites and Spanish-language items. Please add some/any foreign publications;

4) Some typos in the text, e.g. line 595 and other;

5) Some figures are unreadable e.g. 12 and 17;

6) It would be nice to see a table showing the data reduction during the workflow in terms of quantities (e.g. numbers for input data polygons/lines, node count, segments (etc.) before/after aggregation at certain steps in the process, and processing time needed);

7) It would also be nice to see the actual code that was used to produce graphics;

8)  A comparison with other data sources or on-site observations (examples/ground truth) could help to better critically reflect the workflow results and illustrate where the model/workflow resulted in errors, ambiguities, or inaccuracies, either due to data or method used.

We look forward to receiving your revised manuscript.

Kind regards,

Claudionor Ribeiro da Silva

Academic Editor

PLOS ONE

Journal Requirements:

4. We note that Figures 1, 3, 4, 5, 6, 7, 8, 9, 11, 12, 13, 15 and 17 in your submission contain [map/satellite] images which may be copyrighted. All PLOS content is published under the Creative Commons Attribution License (CC BY 4.0), which means that the manuscript, images, and Supporting Information files will be freely available online, and any third party is permitted to access, download, copy, distribute, and use these materials in any way, even commercially, with proper attribution. For these reasons, we cannot publish previously copyrighted maps or satellite images created using proprietary data, such as Google software (Google Maps, Street View, and Earth). For more information, see our copyright guidelines: http://journals.plos.org/plosone/s/licenses-and-copyright.

a. You may seek permission from the original copyright holder of Figures 1, 3, 4, 5, 6, 7, 8, 9, 11, 12, 13, 15 and 17 to publish the content specifically under the CC BY 4.0 license.  

Reviewers' comments:

Reviewer's Responses to Questions

**Comments to the Author**

1. Is the manuscript technically sound, and do the data support the conclusions?

Reviewer #1: Yes

Reviewer #2: Yes

Reviewer #3: Yes

2. Has the statistical analysis been performed appropriately and rigorously? 

Reviewer #1: Yes

Reviewer #2: N/A

Reviewer #3: Yes

3. Have the authors made all data underlying the findings in their manuscript fully available?

Reviewer #1: Yes

Reviewer #2: Yes

Reviewer #3: Yes

4. Is the manuscript presented in an intelligible fashion and written in standard English?

Reviewer #1: Yes

Reviewer #2: Yes

Reviewer #3: Yes

5. Review Comments to the Author

Reviewer #1: I would like to congratulate the author to revised and improve the quality of the manuscript. The manuscript is now written very well and the analysis carried out in this research is new and the results represents the actual study performed.

Reviewer #2: Review of the article “Modeling Barcelona sidewalks: A high resolution urban scale assessment of the geometric attributes of the walkable network”

In this paper authors presents a method of modeling the street network from the perspective of foot traffic, beyond the vehicle-focused street centerline representation approach in transportation research. A scalable method to extract the centerlines of the complete walkable urban area from its polygon representation at a one-meter resolution is discussed, using open-source tools. evaluate the betweenness centrality in a spatial directed graph, the process is applied to the study of the ‘walkable Barcelona’, focusing on three key parameters: 1) the street width, 2) the longitudinal slope, and 3) the crosswalks connecting the sidewalk platforms. The results identify the uneven distribution of these parameters within a complex urban fabric, and the high-resolution cartography allows the identification of critical areas within the network, introducing future lines of research and applications of the model. This is especially relevant considering the increasing awareness of citizens and the urban agendas worldwide, aimed at improving and widening the sidewalk infrastructure that supports local activity in cities

General thoughts

The article presents interesting research and very well prepared. Test procedure is clear and justified, study data are rightly chosen and sufficient, authors correctly described the study. It is important that the test procedure can be reproduced by other researchers. The following detailed comments are intended to correct some of the shortcomings of the article.

Detailed comments

Text is written in a logical and thoughtful way, creating a coherent whole, in accordance with the writing regime of scientific paper (IMRaD). The method of presentation, comprehensive introduction and very interesting and thoughtful practical examples deserve praise. Below a comments for corrections:

1) In my opinion text is too long in some way it might be shorten which will affect on its better reception

2) In the webpages access date needs to be added

3) There are very few scholarly articles cited in the text, and very many websites and Spanish-language items. Please add some/any foreign publications

4) See remark 2, regarding references

5) Do the authors believe that a similar type of research can be applied in geography and tourism to trail research? Please refer into the text adopting below positions:

a) Chwedczuk K, Cienkosz D, Apollo M, et al (2022) Challenges related to the determination of altitudes of mountain peaks presented on cartographic sources. Geod Vestn 66:49–59. https://doi.org/10.15292/geodetski-vestnik.2022.01.49-59

b) Csapó J, Wetzl V (2016) Possibilities for the Creation of Beer Routes in Hungary: A Methodological and Practical Perspective. Eur Countrys 8:250–262. https://doi.org/10.1515/EUCO-2016-0018

c) Mionel V, Mionel O (2016) Cycle tourism in Olt county, Romania. (Re)discovering potential of history and geography for tourism. Amfiteatru Econ 18:913–928

6) Some typos in the text, e.g. line 595 and other

7) Some figures and unreadable e.g. 12 and 17

Text might be accepted for a publication after improvements.

All the best and stay safe

Reviewer #3: General Statement:

The paper demonstrates a detailed implementation workflow for processing urban spatial information to extract sidewalks in Barcelona as a case study. These are used to generate pedestrian network information such as shortest paths, network costs for different edges and segments or summary metrics for different distributions. The results are generally interpret, listing benefits of the implementation and several opportunities for application.

1. The strengths of the paper are its nice graphics and the holistic and good explanations for taking different routes and combining different tools and software along the way, including the descriptions of dead ends. I am no expert in network or topology processing, so I cannot comment on the overall significance or novelty of this papers contributions, particularly for the sub-field of pedestrian network extraction. The short literature review suggests that there is limited comparable work available. That said, the results and process look quite polished, the discussion is reflected and comprehensive, and I would argue that the biggest value for readers lies in the practical implementation and the decision tree that is described in the text. The text is easy to follow and overall well written.

2. Unfortunately, the decision tree for the workflow is not shown as a figure and the practical implementation can only be followed using the text. Figure 2 is a good first overview of the overall workflow, but it is linear, whereas in the manuscript, positively, several dead ends are described, software or data limits, alternative ways or sensitivity checks being done that are not visible in Fig 2. To someone who wants to transfer the workflow to another city, a decision tree would be quite helpful, indicating these options and alternative routes for adaption (e.g. in case of data availability and other changing parameters).

3. It would also be nice to see the actual code that was used to produce graphics. Given that the authors mention Pandas, SciPy, PostGis (etc.) and automatic processing (L.889), I would expect that some kind of Code-Notebook is used (e.g. Jupyter) – perhaps the authors could share these Notebooks or the Code as Supplementing Material. This would also allow readers of the paper to transfer the shown implementation more easily to other cities, which from my point of view would be the most critical contribution.

4. Performance of the data processing appears to play a crucial role in deciding which tools/software or methods were chosen for the various parts of the workflow. It would be nice to see a table showing the data reduction during the workflow in terms of quantities (e.g. numbers for input data polygons/lines, node count, segments (etc.) before/after aggregation at certain steps in the process, and processing time needed).

5. Given the immense data processing being done, I found the discussion of results a bit shallow – I could imagine more comparison with external or existing data sources (e.g. in terms of quality or application), or perhaps a discussion of the impact of this new data on Barcelona city policy/planning – although I understand if this is not possible, given the currently already quite lengthy manuscript. A comparison with other data sources or on-site observations (examples/ground truth) could help to better critically reflect the workflow results and illustrate where the model/workflow resulted in errors, ambiguities, or inaccuracies, either due to data or method used.

I think both of the important suggestions (1. notebooks/code as supplementary materials and 2. workflow decision tree) should be possible with minor modifications or limited additional work. I added a number of minor formal suggestions and a few more specific questions below. I suggest Minor revisions.

Specific Questions:

LL. 477: The medial axis explained here and used later for the network, especially for larger plazas and areas, appears to follow through the center. Wouldn’t pedestrians, who are trying to find the shortest path, walk along the edges of plazas (etc.), instead of walking through plazas based on the nodes located in the middle?

LL. 558: How long were the longest segments? Could they also pass a ridge? (peak in the middle, e.g. where start and end of the segment would be the same height, falsely leading to a 0% slope)

LL. 733: This may be related to my absence of network topology knowledge, but perhaps you could explain why forward and backward directions needed to be analyzed separately. I would expect that a forward weight on a slope uphill is simply inversed on the reverse downhill direction. I wonder why a separate computation is necessary.

L. 866: „The method“ – later on, you use „the model“ (L. 877). I am not picky about these terms. Generally, I would suggest to use ‚workflow‘, since you combine many individual methods in a specific way for the Barcelona case study, and transfer to other cities is not shown. In any case, I would be consistent and clearly distinguish what you mean with model and method, if both terms are used.

Specific comments:

L. 37: Remove “around the world”

L. 81: “[…] by A. Svetsuk (17) [...]” - Remove "A. "

L. 135: Replace “Wien” with Vienna.

L. 430: It should be briefly described what „straight skeleton“ (and other specific network topology terms) means.

L. 564/565: This sentence is difficult to understand. What is the ‘densification step’, and why are two steps necessary instead of tuning parameters of the first step?

L. 595: The equation is not formatted correction/not legible.

L. 587/688: Figure 11 shows nodes, and random colors of nodes, but the weight is not shown – perhaps add a legend for the width of lines with unit of measurement.

L. 717/718: Sentence is difficult to read, there seems to be a missing word between discarded and aggregating.

L. 744: „linestring layer“ seems to be a specific term related to a specific software, perhaps describe more generally.

L. 807/808: “the fact” used twice.

L. 838: This sentence is difficult to understand. Percentage sign missing?

L. 935: “The most used sidewalks” – insert “likely” (e. g.), to indicate that this is not based on observation, but projected data from your workflow

Formal: All Internal references are broken and point to Zotero online links.

6. PLOS authors have the option to publish the peer review history of their article (what does this mean?). If published, this will include your full peer review and any attached files.

Reviewer #1: No

Reviewer #2: No

Reviewer #3: **Yes: **Alexander Dunkel

---

## [Author Response · Author response to Decision Letter 0]

23 Mar 2023

///Dear editor of the Journal,

We have read with interest the comments raised by the reviewers and we have revised the issues identified in their notes and updated the manuscript that we submit in a revised form. The results have also been included as supporting information files.

The authors would like to thank the Academic editors and the Reviewers for the opportunity to address the comments. The authors hope that the Editors and Reviewers will be satisfied with the amendments which we have made to the manuscript after considering the feedback provided.

In relation to the main points indicated at the beginning of the response received, we have corrected or improved the following aspects:

1) Text is too long. If possible try to shorten it without losing quality

///The manuscript text has been made substantially more concise without compromising the explanation of the workflow. The reduction in length has been achieved reducing unnecessary detail in some of the explanations; some of these details are now summarized graphically in a new figure and another that has been updated, and some paragraphs have been removed as it has been considered that excessive implementation details made the readability of the text more difficult. The authors believe that the text is in its current form much easier to follow without losing quality.

2) In the webpages access date needs to be added

///The webpages access dates have been included in the text. We apologize for this oversight.

3) There are very few scholarly articles cited in the text, and very many websites and Spanish-language items. Please add some/any foreign publications

///From our perspective, the paper is based on an introductory discussion on the existing bibliography on the issue and the application of some libraries published in open repositories and websites. The development of the methodology described in the paper does not require of further citation along the central chapters of the paper. However, we have been completed the text with some publications. The paper now has around 64 cited bibliography, 19 of them are Spanish research focusing on the case-study discussion.

4) Some typos in the text, e.g. line 595 and other

///The text has been thoroughly revised and any typos in the manuscript have been corrected. The authors thank the reviewers for pointing this out.

5) Some figures are unreadable e.g. 12 and 17

///The figures have been included again in full resolution and new typographies. Also, we believe the lack of quality might be a consequence of the compression in the content management system. The authors apologize if they are difficult to read. Please note that the figures might have changed numbering as a new figure has been introduced.

6) It would be nice to see a table showing the data reduction during the workflow in terms of quantities (e.g. numbers for input data polygons/lines, node count, segments (etc.) before/after aggregation at certain steps in the process, and processing time needed)

///A new figure has been included in the manuscript that summarizes the changes in geometric complexity in each of the processing steps, and the details regarding the processing time of the workflow. This figure has contributed to significantly reduce the length of the manuscript and consolidate in a single place many details scattered in the text. The authors appreciate the suggestion of the reviewer for pointing this possibility.

7) It would also be nice to see the actual code that was used to produce graphics

///The authors are working on releasing the code in the future with proper documentation and streamline the workflow for external usage. However, the code can be inspected on request.

8) A comparison with other data sources or on-site observations (examples/ground truth) could help to better critically reflect the workflow results and illustrate where the model/workflow resulted in errors, ambiguities, or inaccuracies, either due to data or method used

///The authors agree with the reviewers that a comparison with the ground truth would be interesting to be included, but they considered that it would increase the length of the manuscript excessively. It is however discussed in subsection 2.1.

9) We note that Figures 1, 3, 4, 5, 6, 7, 8, 9, 11, 12, 13, 15 and 17 in your submission contain [map/satellite] images which may be copyrighted. All PLOS content is published under the Creative Commons Attribution License (CC BY 4.0), which means that the manuscript, images, and Supporting Information files will be freely available online, and any third party is permitted to access, download, copy, distribute, and use these materials in any way, even commercially, with proper attribution. For these reasons, we cannot publish previously copyrighted maps or satellite images created using proprietary data, such as Google software (Google Maps, Street View, and Earth). For more information, see our copyright guidelines: http://journals.plos.org/plosone/s/licenses-and-copyright.

///The figures are not using satellite images but base Cartography from CartoBCN open data repository (Barcelona City Council), under CC BY 4.0. This information has been added to each caption.

Reviewers' comments:

Reviewer #1: I would like to congratulate the author to revised and improve the quality of the manuscript. The manuscript is now written very well and the analysis carried out in this research is new and the results represents the actual study performed.

///The authors appreciate the reviewer feedback. The manuscript has been revised to address the issues pointed out by the reviewers.

 

Reviewer #2: (…) The following detailed comments are intended to correct some of the shortcomings of the article.

Detailed comments

Text is written in a logical and thoughtful way, creating a coherent whole, in accordance with the writing regime of scientific paper (IMRaD). The method of presentation, comprehensive introduction and very interesting and thoughtful practical examples deserve praise. Below a comments for corrections:

1) In my opinion text is too long in some way it might be shorten which will affect on its better reception.

////The manuscript text has been made substantially more concise without compromising the explanation of the workflow. The reduction in length has been achieved reducing unnecessary detail in some of the explanations; some of these details are now summarized graphically in a new figure and another that has been updated, and some paragraphs have been removed as it has been considered that excessive implementation details made the readability of the text more difficult. The authors believe that the text is in its current form much easier to follow without losing quality.

2) In the webpages access date needs to be added

///This information has been added to the text. The authors thank the reviewer for pointing this out.

3) There are very few scholarly articles cited in the text, and very many websites and Spanish-language items. Please add some/any foreign publications

///From our perspective, the paper is based on an introductory discussion on the existing bibliography on the issue and the application of some libraries published in open repositories and websites. The development of the methodology described in the paper does not requires of further citation along the central chapters of the paper. However, we have been completed the text with some publications. The paper now has around 64 cited bibliography, 19 of them are Spanish research focusing on the case-study discussion.

4) See remark 2, regarding references

This information has been added to the text.

5) Do the authors believe that a similar type of research can be applied in geography and tourism to trail research? Please refer into the text adopting below positions:

a) Chwedczuk K, Cienkosz D, Apollo M, et al (2022) Challenges related to the determination of altitudes of mountain peaks presented on cartographic sources. Geod Vestn 66:49–59. https://doi.org/10.15292/geodetski-vestnik.2022.01.49-59

b) Csapó J, Wetzl V (2016) Possibilities for the Creation of Beer Routes in Hungary: A Methodological and Practical Perspective. Eur Countrys 8:250–262. https://doi.org/10.1515/EUCO-2016-0018

c) Mionel V, Mionel O (2016) Cycle tourism in Olt county, Romania. (Re)discovering potential of history and geography for tourism. Amfiteatru Econ 18:913–928

///We do thank very much the reviewer for these key references that might be related to the topic of the paper. The methodology is applied to sidewalks as polygon geometries with specific attributes. However, further research might be helpful to find other utilities in other fields of knowledge such as geography. Unfortunately, in order to make the paper more concise, we consider that this discussion should be addressed in future research.

6) Some typos in the text, e.g. line 595 and other

///The text has been fully reviewed. We apologize for this oversight.

7) Some figures and unreadable e.g. 12 and 17

///The figures have been included again in full resolution and new typographies. Also, we believe the lack of quality might be a consequence of the compression in the content management system. The authors apologize if they are difficult to read. Please note that the figures might have changed numbering as a new figure has been introduced.

 

Reviewer #3: General Statement:

///The paper demonstrates a detailed implementation workflow for processing urban spatial information to extract sidewalks in Barcelona as a case study. These are used to generate pedestrian network information such as shortest paths, network costs for different edges and segments or summary metrics for different distributions. The results are generally interpret, listing benefits of the implementation and several opportunities for application.

1. The strengths of the paper are its nice graphics and the holistic and good explanations for taking different routes and combining different tools and software along the way, including the descriptions of dead ends. I am no expert in network or topology processing, so I cannot comment on the overall significance or novelty of this papers contributions, particularly for the sub-field of pedestrian network extraction. The short literature review suggests that there is limited comparable work available. That said, the results and process look quite polished, the discussion is reflected and comprehensive, and I would argue that the biggest value for readers lies in the practical implementation and the decision tree that is described in the text. The text is easy to follow and overall well written.

///The authors appreciate the reviewer feedback. The manuscript has been revised to address the issues pointed out by the reviewers.

2. Unfortunately, the decision tree for the workflow is not shown as a figure and the practical implementation can only be followed using the text. Figure 2 is a good first overview of the overall workflow, but it is linear, whereas in the manuscript, positively, several dead ends are described, software or data limits, alternative ways or sensitivity checks being done that are not visible in Fig 2. To someone who wants to transfer the workflow to another city, a decision tree would be quite helpful, indicating these options and alternative routes for adaption (e.g. in case of data availability and other changing parameters).

///These details are now summarized graphically in a new figure (Fig. 3) with the workflow steps to produce a directed graph from the source data. The complexity of the data is expressed as node and vertex count at each step, in a logarithmic scale. The processing time required is shown as elapsed minutes from the initial condition.

Another figure (currently Fig. 2 in the manuscript) that has been updated, where the processing step appears in bold with the selected processing tool in parentheses below. The main motivation of the tool choice among alternatives appears in italics underneath. The workflow is indeed linear, because dead ends in the workflow where discarded and not pursued further, and therefore the current figure depicts a decision tree with a single branch.

3. It would also be nice to see the actual code that was used to produce graphics. Given that the authors mention Pandas, SciPy, PostGis (etc.) and automatic processing (L.889), I would expect that some kind of Code-Notebook is used (e.g. Jupyter) – perhaps the authors could share these Notebooks or the Code as Supplementing Material. This would also allow readers of the paper to transfer the shown implementation more easily to other cities, which from my point of view would be the most critical contribution.

////The authors are working on releasing the code in the future with proper documentation and streamline the workflow for external usage, as in its current form it is hard coded to be processed in a specific database structure. However, the code can be inspected on request.

4. Performance of the data processing appears to play a crucial role in deciding which tools/software or methods were chosen for the various parts of the workflow. It would be nice to see a table showing the data reduction during the workflow in terms of quantities (e.g. numbers for input data polygons/lines, node count, segments (etc.) before/after aggregation at certain steps in the process, and processing time needed).

///These details are now summarized graphically in a new figure (Fig. 3. The complexity of the data is expressed as node and vertex counts at each step, in a logarithmic of base-2 scale, indicating a doubling or halving of the complexity at each step. The processing time required is shown as elapsed minutes from the initial condition.

5. Given the immense data processing being done, I found the discussion of results a bit shallow – I could imagine more comparison with external or existing data sources (e.g. in terms of quality or application), or perhaps a discussion of the impact of this new data on Barcelona city policy/planning – although I understand if this is not possible, given the currently already quite lengthy manuscript. A comparison with other data sources or on-site observations (examples/ground truth) could help to better critically reflect the workflow results and illustrate where the model/workflow resulted in errors, ambiguities, or inaccuracies, either due to data or method used.

///The authors thank the reviewer for pointing it out. We believe that including this discussion in the manuscript in its current form, even after being made more concise, would make it too lengthy. Because the focus of the manuscript is on the workflow, the authors plan to address these and other aspects in future work.

I think both of the important suggestions (1. notebooks/code as supplementary materials and 2. workflow decision tree) should be possible with minor modifications or limited additional work. I added a number of minor formal suggestions and a few more specific questions below. I suggest Minor revisions.

Specific Questions:

LL. 477: The medial axis explained here and used later for the network, especially for larger plazas and areas, appears to follow through the center. Wouldn’t pedestrians, who are trying to find the shortest path, walk along the edges of plazas (etc.), instead of walking through plazas based on the nodes located in the middle?

///The medial axis has been considered as the most representative of the actual trajectories of pedestrians. Avoiding moving obstacles such as other pedestrians, as well as perceptual cues or momentum in curved trajectories can also play a role. The medial axis tries to simplify these lines to the average trajectory as a summary of all possibilities. While we agree that it is true that a very detailed inspection inevitably reveal issues, at the scale of analysis these issues are minor as they are aggregated into a single summary. 

LL. 558: How long were the longest segments? Could they also pass a ridge? (peak in the middle, e.g. where start and end of the segment would be the same height, falsely leading to a 0% slope)

///The slope was evaluated along all the length of the segments, draping the vertices to the topographical surface. The simplification step included a parameter to preserve detail that was set to 0.5m. Each of the individual segments between each pair of vertices along the linestring had their slope computed separately, and converted to a weight in time units according to their length and slope using Tobler's formula. The sum of these time units was aggregated in the simplified network, as slope could not be aggregated because its effects are non-linear.

LL. 733: This may be related to my absence of network topology knowledge, but perhaps you could explain why forward and backward directions needed to be analyzed separately. I would expect that a forward weight on a slope uphill is simply inversed on the reverse downhill direction. I wonder why a separate computation is necessary.

///The time to transverse a sloped terrain depends on the direction of travel, as it is harder to climb uphill than to descend downhill. Therefore, it must be considered in what direction the simulated flow is traveling to assign the corresponding weight. Since it depends on the origins and destinations in each o*d calculation, it cannot be set in advance because sometimes it can be transversed in one direction and in other cases in the opposite. This is the reason why a directed graph with different weights in each direction was essential, and in our opinion one of the most overlooked aspects in these analyses, which we plan to address further in future work.

L. 866: „The method“ – later on, you use „the model“ (L. 877). I am not picky about these terms. Generally, I would suggest to use ‚workflow‘, since you combine many individual methods in a specific way for the Barcelona case study, and transfer to other cities is not shown. In any case, I would be consistent and clearly distinguish what you mean with model and method, if both terms are used.

///The choice of terms has been corrected in the context where it was applicable. The authors thank the reviewer for the suggestion.

Specific comments:

L. 37: Remove “around the world”

L. 81: “[…] by A. Svetsuk (17) [...]” - Remove "A. "

L. 135: Replace “Wien” with Vienna.

L. 430: It should be briefly described what „straight skeleton“ (and other specific network topology terms) means.

L. 564/565: This sentence is difficult to understand. What is the ‘densification step’, and why are two steps necessary instead of tuning parameters of the first step?

L. 595: The equation is not formatted correction/not legible.

L. 587/688: Figure 11 shows nodes, and random colors of nodes, but the weight is not shown – perhaps add a legend for the width of lines with unit of measurement.

L. 717/718: Sentence is difficult to read, there seems to be a missing word between discarded and aggregating.

L. 744: „linestring layer“ seems to be a specific term related to a specific software, perhaps describe more generally.

L. 807/808: “the fact” used twice.

L. 838: This sentence is difficult to understand. Percentage sign missing?

L. 935: “The most used sidewalks” – insert “likely” (e. g.), to indicate that this is not based on observation, but projected data from your workflow

Formal: All Internal references are broken and point to Zotero online links.

///All these minor issues have been thoroughly reviewed and fixed. The authors appreciate the reviewer pointing it out, an apologize for the oversight.

---

## [Decision Letter · Decision Letter 1]

5 Apr 2023

Modeling Barcelona sidewalks: A high resolution urban scale assessment of the geometric attributes of the walkable network

PONE-D-22-35314R1

Dear Dr. Clua,

We’re pleased to inform you that your manuscript has been judged scientifically suitable for publication and will be formally accepted for publication once it meets all outstanding technical requirements.

Kind regards,

Claudionor Ribeiro da Silva

Academic Editor

PLOS ONE

Additional Editor Comments (optional):

Reviewers' comments:

Reviewer's Responses to Questions

**Comments to the Author**

1. If the authors have adequately addressed your comments raised in a previous round of review and you feel that this manuscript is now acceptable for publication, you may indicate that here to bypass the “Comments to the Author” section, enter your conflict of interest statement in the “Confidential to Editor” section, and submit your "Accept" recommendation.

Reviewer #2: All comments have been addressed

Reviewer #3: All comments have been addressed

2. Is the manuscript technically sound, and do the data support the conclusions?

Reviewer #2: Yes

Reviewer #3: Yes

3. Has the statistical analysis been performed appropriately and rigorously? 

Reviewer #2: Yes

Reviewer #3: Yes

4. Have the authors made all data underlying the findings in their manuscript fully available?

Reviewer #2: Yes

Reviewer #3: Yes

5. Is the manuscript presented in an intelligible fashion and written in standard English?

Reviewer #2: Yes

Reviewer #3: Yes

6. Review Comments to the Author

Reviewer #2: (No Response)

Reviewer #3: The new Figure 3 turned out very good for summarizing data reduction along the workflow

I appreciate the authors answers to my previous questions.

I think the paper can be published in its current form. I still think that publishing the code alongside would be critical, particularly since you intent publish in PLoS and as the process is otherwise hard to replicate, but leave this decision to the editors. There are some remaining typos which I have listed below.

LL. 327: what is “the plos geometry”?

LL. 363 “[…] because include the information […]”: seems like a word is missing

LL. 524 “[…] this degree 2 nodes to […]” something seems wrong with this sentence, what does “2” mean here?

7. PLOS authors have the option to publish the peer review history of their article (what does this mean?). If published, this will include your full peer review and any attached files.

Reviewer #2: No

Reviewer #3: **Yes: **Alexander Dunkel

---

## [Editor Report · Acceptance letter]

14 Apr 2023

PONE-D-22-35314R1 

Modeling Barcelona sidewalks: A high resolution urban scale assessment of the geometric attributes of the walkable network 

Dear Dr. Clua:

I'm pleased to inform you that your manuscript has been deemed suitable for publication in PLOS ONE. Congratulations! Your manuscript is now with our production department. 

Kind regards, 

on behalf of

Dr. Claudionor Ribeiro da Silva 

Academic Editor

PLOS ONE